# EMC rectifies the topology of multipass membrane proteins

Haoxi Wu ⬡ , Luka Smalinskaitė & Ramanujan S. Hegde ⬡ ✉

Most eukaryotic multipass membrane proteins are inserted into the membrane of the endoplasmic reticulum. Their transmembrane domains (TMDs) are thought to be inserted co-translationally as they emerge from a membrane-bound ribosome. Here we find that TMDs near the carboxyl terminus of mammalian multipass proteins are inserted post-translationally by the endoplasmic reticulum membrane protein complex (EMC). Site-specific crosslinking shows that the EMC's cytosol-facing hydrophilic vestibule is adjacent to a pre-translocated C-terminal tail. EMC-mediated insertion is mostly agnostic to TMD hydrophobicity, favored for short uncharged C-tails and stimulated by a preceding unassembled TMD bundle. Thus, multipass membrane proteins can be released by the ribosome–translocon complex in an incompletely inserted state, requiring a separate EMC-mediated post-translational insertion step to rectify their topology, complete biogenesis and evade quality control. This sequential co-translational and post-translational mechanism may apply to ~250 diverse multipass proteins, including subunits of the pentameric ion channel family that are crucial for neurotransmission.

Multipass membrane proteins, defined by the presence of more than one TMD, play crucial roles in the transfer of information and molecules across biological membranes[1,2]. Most eukaryotic multipass membrane proteins are inserted co-translationally at the endoplasmic reticulum as individual TMDs emerge from a membrane-bound ribosome[3]. Recent work suggests that at the mammalian endoplasmic reticulum, different parts of a multipass protein are inserted by different factors. The first TMD can be inserted by the EMC or by passing through a lateral gate in the Sec61 translocation channel[4–7]. Subsequent TMDs can also be inserted through the Sec61 lateral gate or into a lipid-filled cavity behind Sec61 created by the multipass translocon (MPT), an assembly of three complexes termed PAT, GEL and BOS[8–10]. GEL is structurally and evolutionarily related to the EMC, suggesting that it may be the insertion factor within the MPT[11–13].

It is thought that both EMC and MPT insert TMDs only when the flanking translocated domain is shorter than ~50 amino acids, whereas Sec61 can mediate TMD insertion flanked by longer translocated domains[4,8,9,14]. Sec61 can translocate long hydrophilic polypeptides because it houses an aqueous translocation channel[15–18], a feature lacking in either EMC or any of the MPT complexes[8,19–23]. By using EMC,

Sec61 and MPT in these ways, membrane proteins of widely varying topology and translocated domains can be co-translationally weaved into the lipid bilayer[3,9]. However, the last TMD poses unique problems if it is located within ~50 amino acids of the C terminus (hereafter termed a terminal TMD). As termination occurs before it can engage either Sec61 or the MPT, its insertion is necessarily post-translational. How a terminal TMD of a multipass protein is inserted is not clear.

The most straightforward case is one in which the penultimate TMD and the final TMD are separated by a long translocated loop. In this instance, the translocated loop would already be threaded into the ribosome-bound Sec61 channel at the time of termination (Fig. 1a). The terminal TMD would then necessarily enter the Sec61 channel after termination, from where it would enter the membrane post-translationally, presumably through Sec61's lateral gate. By contrast, the insertion mechanism for a terminal TMD preceded by a short translocated loop (Fig. 1b) or for a terminal TMD followed by a short translocated tail (Fig. 1c) is not clear. In the first situation, the penultimate TMD would not have enough of a tether to have engaged Sec61 before termination. The final two TMDs would be released from the ribosome and both would need to insert post-translationally

MRC Laboratory of Molecular Biology, Cambridge, UK. ✉e-mail: rhegde@mrc-lmb.cam.ac.uk

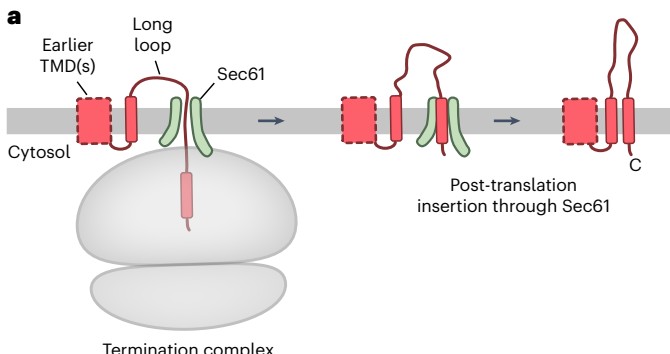

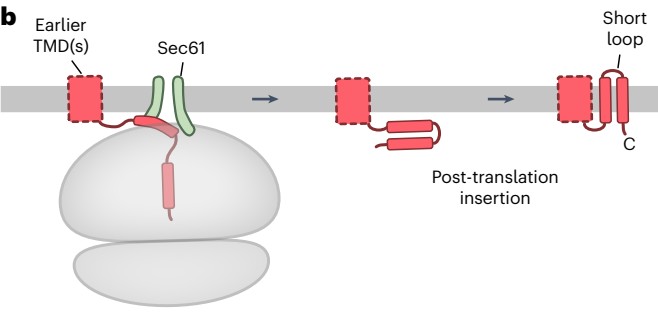

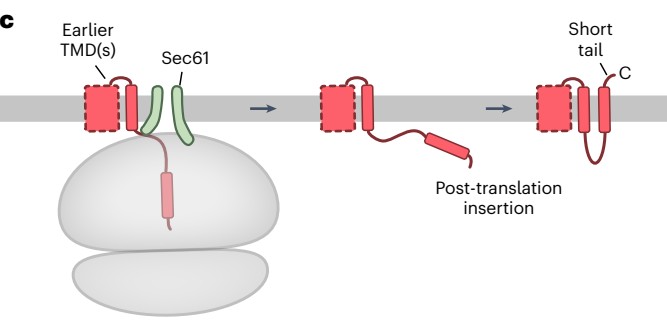

**Fig. 1 | Three modes of terminal TMD insertion for multipass membrane proteins.** TMDs located within ~50 amino acids of the stop codon of a multipass membrane protein will be partially or completely inside the ribosome exit tunnel at the time of translation termination. This means that they will necessarily be inserted by one of three post-translational mechanisms depending on the preceding membrane domain and intervening loop. **a**, When the penultimate TMD is followed by a long (more than 50 amino acids) translocated loop, its translocation will be Sec61-dependent[8]. Hence, the loop will already be threaded through the Sec61 channel by the time of termination (left diagram). The terminal TMD inside the ribosome will then necessarily enter Sec61 (middle diagram), from where it presumably accesses the membrane through Sec61's lateral gate. **b**, When the final two TMDs are closely spaced, neither one can be inserted co-translationally because the tether downstream of the penultimate TMD is too short to engage Sec61 at the time of termination (left diagram). Therefore, both TMDs will be released and inserted post-translationally by an unknown mechanism. **c**, When the final TMD is located near a translocated C terminus, the TMD and tail are mostly or entirely inside the ribosome at the time of termination (left diagram). Their post-translational insertion mechanism is also unknown. Note that the machinery involved in the insertion and chaperoning of earlier TMDs is not shown for simplicity.

concomitant with translocation of the intervening loop. In the second situation, the terminal TMD would also need to be inserted post-translationally concomitant with translocation of the C-terminal tail (Fig. 1c). The factors that mediate these post-translational insertion reactions are not known.

This problem is exemplified by the large and important family of Cys-loop pentameric ion channels. This family includes acetylcholine receptors, γ-aminobutyric acid type A (GABA$_A$) receptors, glycine receptors and others[24]. They play crucial roles in neurotransmission, and their incorrect biogenesis would lead to complex neurologic consequences[25]. Each subunit is composed of four TMDs with both the N terminus and C terminus facing the extracellular environment. The last TMD is followed by a tail that is typically only ~10–20 amino acids. This means that TMD4 will be mostly or entirely within the ribosome when translation terminates. The preceding three TMDs would have already been inserted given the ~100 amino acid long cytosolic loop between TMD3 and TMD4. Hence, TMD4 would be released from the ribosome and must be inserted post-translationally by the route shown in Fig. 1c. Given the exceptional importance of this class of proteins, we investigated how the final TMD of a GABA$_A$ receptor subunit is inserted as a model for the general problem of terminal TMD insertion outlined above.

## Results

### The C-terminal TMD of GABRA1 is inserted by EMC

In considering factors that might mediate terminal TMD insertion of GABA$_A$ receptor subunits, we were intrigued by the earlier observation that the loss of EMC impairs the expression of multiple members of Cys-loop pentameric ion channels in worms[26]. As subunits of these channels use an N-terminal signal sequence for endoplasmic reticulum targeting and initiation of N-terminal translocation, it is now appreciated that insertion of the first TMD is not expected to be EMC-dependent[4]. TMD2 and TMD3 would then insert via the MPT given the short translocated loop between them[8,10]. This suggests that the EMC requirement might be at a later stage of insertion, folding or assembly. Among these possibilities, a potential role in the insertion of TMD4 was attractive because the biochemical reaction is similar to the post-translational insertion of tail-anchored proteins, which is an established role for EMC[7,14].

To examine a potential role for EMC in Cys-loop channels, we focused on GABA$_A$ receptors, whose reliance on EMC has also been seen in mammals[27]. Using a previously characterized HEK293 cell line with stable inducible expression of a heteropentameric GABA$_A$ receptor[28,29], we tested the effect of acute EMC depletion. As seen in earlier studies, induced surface expression of the GABA$_A$ receptor was reduced to less than ~50% in cells knocked down for EMC4, which generally phenocopies the insertase deficiency seen with a loss of EMC2, EMC3, EMC5 or EMC6, other core subunits of EMC[30–32] (Fig. 2a). No effect was seen for a dual-color fluorescent reporter of the Sec61-inserted asialoglycoprotein receptor 1 (ASGR1), but a strong reduction was seen for a similar reporter of the known EMC-inserted tail-anchored protein squalene synthase (SQS). Immunoblotting for GABRA1, the α1 subunit of the GABA$_A$ receptor, indicated that its lack of surface expression corresponded to its degradation, consistent with a failure in biogenesis (Fig. 2b). Importantly, cells lacking EMC contain the normal complement of all major factors involved in endoplasmic reticulum protein biogenesis (Extended Data Fig. 1 and ref. 4), consistent with normal biogenesis of most secretory pathway proteins[4,14,31–33]. This suggests that the effect of EMC on GABA$_A$ receptor biogenesis is unlikely to be indirect.

To examine whether the EMC-dependent step involves membrane insertion of a GABA$_A$ receptor, we reconstituted the insertion of GABRA1 in vitro. Translation of $^{35}$S-labeled GABRA1 in reticulocyte lysate in the presence of semi-permeabilized cells (SPCs) resulted in signal sequence cleavage and glycosylation of the N-terminal domain in the endoplasmic reticulum lumen (Fig. 2c). Identical results were seen using SPCs from EMC knockout (ΔEMC) cells, consistent with previous work showing that signal sequences use Sec61 to initiate translocation of the N-terminal domain independent of EMC[4].

We used a protease protection assay to monitor TMD insertion of GABRA1 downstream of N-terminus translocation. Proteinase K cleaved

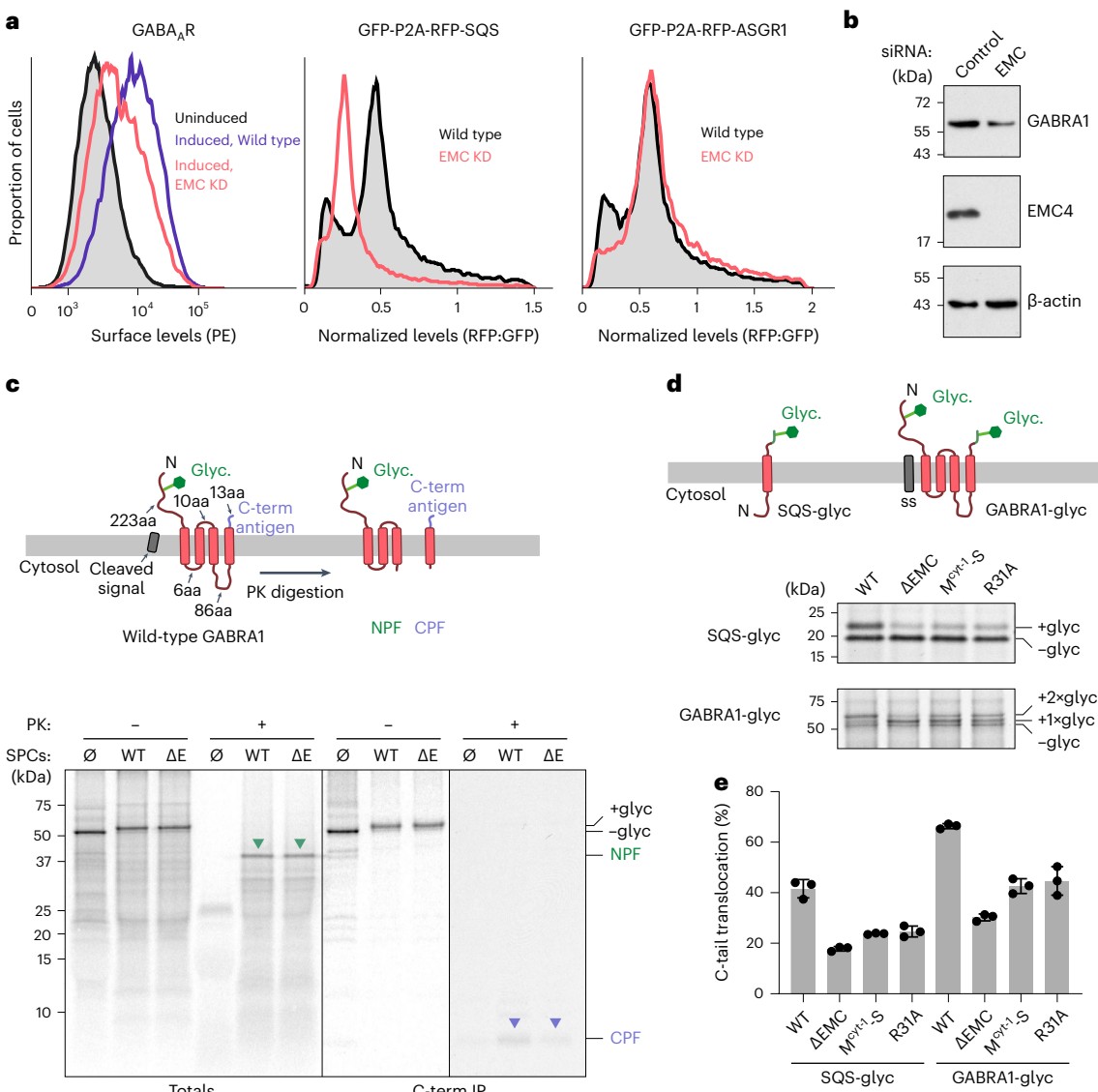

**Fig. 2 | EMC is required for C-terminal TMD insertion of GABRA1. a**, Stable cell lines expressing the indicated inducible constructs were treated with non-targeting or EMC4-targeting siRNAs for 3 days, induced for 6 h and analyzed by flow cytometry for surface GABA$_A$ receptor levels using phycoerythrin (PE) labeled antibody (left), total levels of RFP-SQS (middle) or total levels of RFP-ASGR1 (right). SQS and ASGR1 levels are normalized to GFP, an internal expression control that is separated from the reporter by a ribosome-skipping viral P2A sequence. KD indicates knockdown. **b**, Total levels of GABRA1 and EMC4 were analyzed by blotting in cells treated with control or EMC4 siRNA. β-actin was used as a loading control. **c**, Topology diagram of human GABRA1; the predicted fragments resulting from proteinase K (PK) digestion are shown on top. Tail and loop lengths are indicated. NPF, N-terminal protected fragment; CPF, C-terminal protected fragment. The bottom panel shows the topology analysis of GABRA1 in the endoplasmic reticulum of wild-type (WT) and EMC6-knockout (ΔE) 293 cells. $^{35}$S-methionine-labeled GABRA1 was translated in rabbit reticulocyte lysate in the absence (Ø) or presence of SPCs derived from WT or ΔE 293 cells. After

translation, the SPCs were recovered by centrifugation and analyzed directly (−PK) by SDS−PAGE and autoradiography or subjected to PK digestion (+PK). Reactions lacking SPCs were analyzed similarly without centrifugation. Aliquots of both −PK and +PK samples were subjected to immunoprecipitation via an antibody against the C terminus of GABRA1 (antigen labeled in purple). The glycosylated (+glyc) and non-glycosylated (−glyc) products are indicated. NPF and CPF are indicated with green and purple arrowheads, respectively. **d**, $^{35}$S-methionine-labeled SQS and GABRA1 each with a C-terminal glycosylation site were translated in the presence of SPCs derived from ΔEMC cells or cells stably overexpressing either wild-type EMC3-FLAG or insertase-deficient mutants of EMC3-FLAG variants (M$^{cyt-1}$-S and R31A). SS indicates a cleavable signal sequence. **e**, Quantification of three independent experiments as shown in **d**, with mean ± s.d. plotted. C-tail translocation was quantified by plotting per cent glycosylation (for SQS) or the per cent of glycosylated products that contain two glycans (For GABRA1). C-term, C-terminal.

the long cytosolic loop between TMD3 and TMD4, generating two protected fragments: an N-terminal protected fragment corresponding to the first three TMDs of GABRA1 and a C-terminal protected fragment corresponding to TMD4 and a short translocated C-terminal tail (Fig. 2c). Wild-type and ΔEMC SPCs generate similar levels of N-terminal protected fragments after proteinase K digestion, indicating that EMC does not participate in the insertion of the first three TMDs of GABRA1.

By contrast, The C-terminal protected fragment (recovered using an antibody against the C-tail of GABRA1) is reduced in ΔEMC SPCs to less than half that seen in wild-type SPCs, suggesting an impairment in TMD4 insertion. To corroborate this conclusion, we introduced a glycosylation site into the GABRA1 C-tail (GABRA1-glyc) to monitor C-tail translocation as a proxy for TMD4 insertion (Fig. 2d). Glycosylation of the C-tail was impaired by more than 50% in the absence of

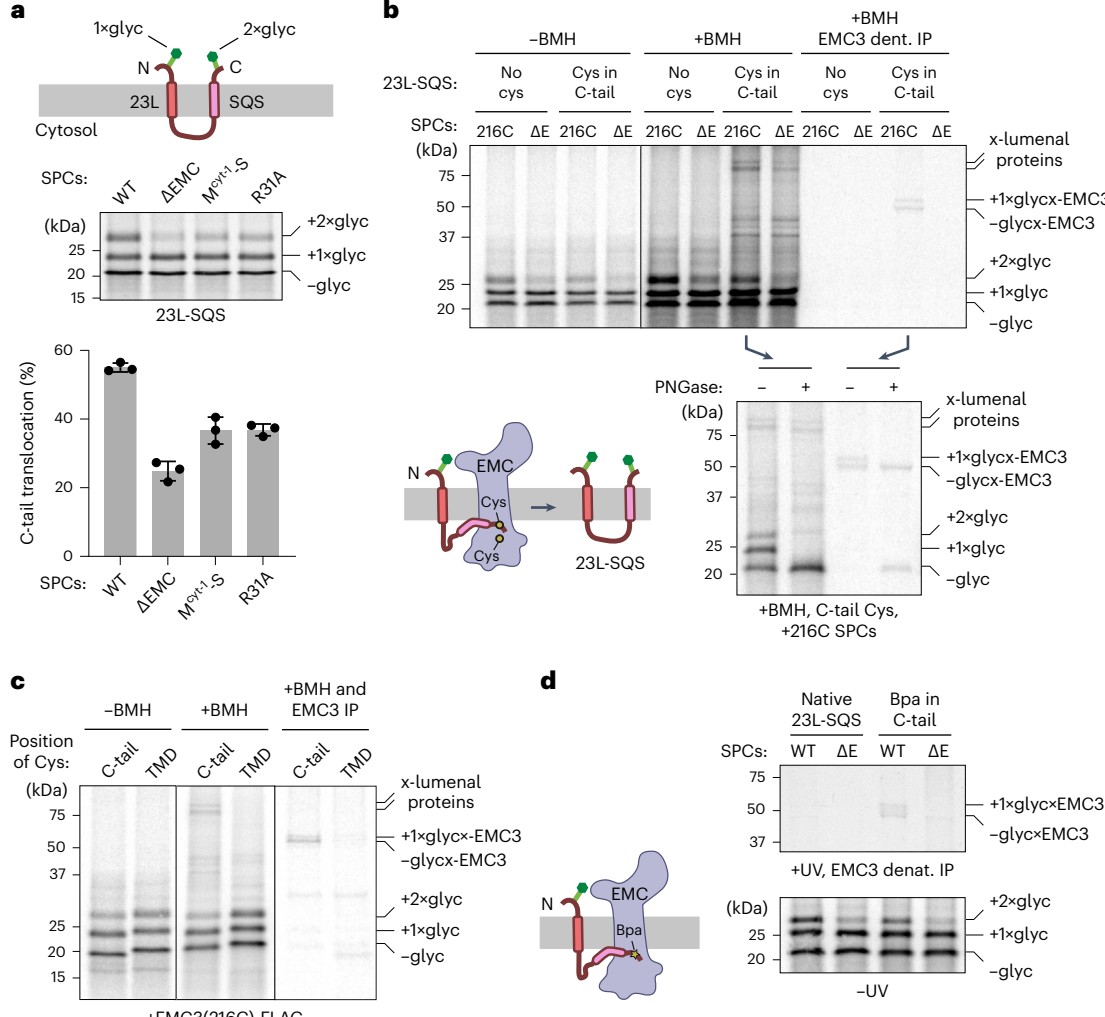

**Fig. 3 | Substrate C-tail samples EMC before translocation. a**, The 23L-SQS reporter (top) consists of a short translocated N-tail, a first TMD made of 23 leucine residues, an ~100 amino acid cytosolic loop, a second TMD and flanking sequences from SQS and a short translocated C-tail. Both terminal tails have glycosylation sites to monitor translocation. [35]S-methionine labeled 23L-SQS was translated in the presence of SPCs derived from ΔEMC cells or cells expressing variants of EMC3 (WT, M[cyt-1]-S and R31A). Products with different glycosylation states are indicated. Quantification of three independent experiments (mean ± s.d.) is plotted. C-tail translocation is determined as the per cent of glycosylated products that contains two glycans. **b**, [35]S-methionine-labeled 23L-SQS without or with a single cysteine within the C-tail, translated in the presence of SPCs derived from ΔEMC (ΔE) cells or cells expressing EMC3-216C (in which a single cysteine is introduced into cysteine-free EMC3 at residue 216 located in its cytosolic vestibule)[6]. One aliquot was analyzed directly (−BMH) and another was treated with BMH. Crosslinked samples were analyzed directly

(+BMH) or after EMC3 denaturing immunoprecipitation via FLAG tag (+BMH EMC3 denat. IP). The positions of 23L-SQS with zero, one or two glycans, and crosslinks between 23L-SQS and EMC3 or lumenal proteins are indicated. Lanes containing EMC3 crosslinks were digested with PNGase F to confirm that the crosslinks contain either zero or one glycan and hence are not crosslinks to fully translocated double-glycosylated 23L-SQS. **c**, [35]S-methionine-labeled 23L-SQS containing a single cysteine within the C-tail or the SQS TMD, translated in the presence of EMC3-216C SPCs and analyzed similarly to **b**. **d**, [35]S-methionine-labeled 23L-SQS without or with a single Bpa incorporated within the C-tail (through an amber suppression system) was translated in the presence of SPCs derived from WT or ΔE cells. One aliquot was analyzed directly (−UV) and another was subjected to UV crosslinking. Crosslinked samples were analyzed after EMC3 denaturing immunoprecipitation via FLAG tag (+UV EMC3 denat. IP). Labeling is similar to **b**.

EMC. This level of impairment was similar to that seen for SQS-glyc, a well-established substrate for EMC-mediated insertion[14].

As with SQS, insertion of GABRA1 TMD4 was not entirely eliminated in ΔEMC SPCs, presumably explaining why GABA_A receptor surface expression is not completely eliminated in ΔEMC cells. We considered whether the residual GABRA1 TMD4 insertion in ΔEMC SPCs could potentially be mediated by the lateral gate of Sec61, the GEL complex or the GET complex, which inserts tail-anchored proteins of high hydrophobicity. Of these, the GET insertase seemed unlikely because later crosslinking experiments did not observe an interaction between a membrane-tethered C-terminal TMD and GET3, the targeting factor required for access to the GET insertase[14,34,35]. As the

Sec61 and GEL complexes are probably used for GABRA1 N-terminal translocation and insertion of the TMD2–TMD3 module, respectively, we devised a simplified reporter to test TMD4 insertion in isolation. This C-tail translocation reporter consists of an artificial signal-anchor comprising 23 leucine residues (23L) followed by a cytosolic loop, TMD4 of GABRA1 and a translocated C-tail (23L-GABRA1). As 23L can be inserted independently of EMC, GEL or Sec61, we can test the role of these factors in TMD4 insertion. Glycosylation sites in the N-terminal and C-terminal tails were used to monitor translocation.

As with full-length GABRA1, C-tail translocation of 23L-GABRA1 was EMC-dependent. Elimination of the GEL complex (by knockout of its TMCO1 subunit) or treatment with Apratoxin A (ApraA), a potent

inhibitor of Sec61's lateral gate[36–38], had no effect on 23L-GABRA1 insertion (Extended Data Fig. 2). Combining one or both manipulations with the elimination of EMC showed no further impairment of C-tail translocation, suggesting that neither Sec61 nor GEL can contribute to TMD4 insertion. These results indicate that TMD4 insertion is primarily mediated by EMC, with residual EMC-independent insertion occurring unassisted or through an unknown insertase. Given the experimental and theoretical support for unassisted insertion[39,40], a membrane-tethered TMD4 could readily access this insertion route. Nonetheless, such alternative mechanisms seem to be minor contributors relative to EMC.

EMC uses a cytosol-facing hydrophilic vestibule and membrane-embedded hydrophilic groove, both housed primarily in EMC3, to facilitate translocation of a flanking hydrophilic segment concomitant with TMD insertion[7,19,21,23]. To test whether this established mechanism was used for TMD4 insertion of GABRA1, we analyzed the effect of EMC3 mutations along the translocation route. In these experiments, ~70–90% of endogenous EMC3 is replaced by long-term stable overexpression of FLAG-tagged EMC3, with the excess EMC3 being effectively degraded by cellular quality control[6]. Mutation of EMC3 at either a cytosolic methionine-rich loop at the entry of the hydrophilic vestibule ($M^{cyt-1}$-S) or a charged residue within the hydrophilic groove (R31A) impaired insertion of both SQS and TMD4 of GABRA1 (Fig. 2e). This result shows that EMC's insertase activity is involved in terminal TMD insertion, not some other part of EMC such as its putative chaperone surface[41]. We infer this conclusion because the degree of impairment in these mutants is similar for both SQS and TMD4 of GABRA1, and because these EMC mutants are fully assembled and intact[23]. Thus, the last TMD of GABRA1 is post-translationally inserted into the endoplasmic reticulum membrane through EMC using a similar mechanism as previously known substrates.

### Translocating C-tail samples EMC during insertion

The 23L-TMD strategy provided a simplified reporter to analyze substrate parameters that influence EMC-dependent terminal TMD insertion. However, GABRA1 TMD4 insertion efficiency in this reporter was relatively low (a point we will return to later), so we devised a more efficiently inserted construct containing the TMD of SQS (23L-SQS; Fig. 3a). The linker between 23L and SQS was of the same length (~100 amino acids) as the cytosolic loop preceding the final TMD of GABRA1, so the 23L domain would have been inserted and diffused away from the translocon by the time translation terminates and the terminal TMD emerges from the ribosome[42]. We confirmed that 23L-SQS retains EMC-dependent insertion of the SQS TMD, which was impaired by mutations in EMC's insertase path (Fig. 3a).

With this construct, we used site-specific chemical crosslinking to probe the local environment during C-terminal TMD insertion. A single cysteine in the C-tail of 23L-SQS crosslinked to a single cysteine engineered in the cytosolic vestibule of EMC at position 216 of EMC3 (Fig. 3b). EMC3 crosslinked to non-glycosylated and singly glycosylated substrate, but not to doubly glycosylated substrate. This indicates that EMC's cytosolic vestibule can crosslink to a substrate whose C-terminal TMD has yet to be inserted but whose N-terminal TMD has already been inserted (and is glycosylated). Importantly, a cysteine in the middle of the adjacent SQS TMD showed almost no crosslinking to EMC3 (Fig. 3c). This provides a specificity control for the observed tail-mediated crosslinks and supports a model in which EMC mediates translocation of the hydrophilic tail through its hydrophilic vestibule.

In further support of this idea, two independent EMC3 mutants that partially impair translocation led to a ~1.9-fold increase in crosslinks between the C-tail and EMC's cytosolic vestibule (Extended Data Fig. 3). This observation is consistent with a longer residence time at this pre-translocation step when the translocation reaction is impaired, similar to earlier observations for an N-terminal EMC-dependent TMD[6]. Although less efficient, a UV-activated photo-crosslinker in the C-tail of

23L-SQS crosslinked to EMC3 in wild-type EMC (Fig. 3d). These results indicate that an incompletely inserted membrane protein encounters EMC, presumably by diffusional sampling within the membrane, and uses EMC's hydrophilic vestibule for translocation of the C-terminal tail concomitant with TMD insertion. In the absence of a functional EMC insertase, the C-terminal TMD presumably binds rapidly to the nearby membrane surface[40] but evidently cannot be inserted as efficiently by an unassisted mechanism or by another insertion factor (Extended Data Fig. 2). Binding to the membrane surface might explain why the TMD does not show any obvious increase in crosslinking to cytosolic chaperones or targeting factors in the absence of EMC as might otherwise be expected for a TMD exposed to the cytosol[14,43–45].

### Determinants of EMC-mediated C-terminal TMD insertion

The 23L-SQS reporter was then modified to test the substrate features that influence terminal TMD insertion. In the first series of experiments, we found that extending the C-tail shifted the insertion pathway from EMC to Sec61 (Fig. 4a). At lengths of 25 and 35 amino acids, the TMD is insufficiently exposed outside the ribosome to engage the Sec61 lateral gate in the hairpin topology required for C-tail translocation. These constructs were, therefore, unaffected by the Sec61 inhibitor ApraA and were sensitive to the loss of EMC. By contrast, at 55 amino acids or longer, C-tail translocation was unaffected in ΔEMC SPCs but completely inhibited by ApraA. The 45 amino acid tail showed an intermediate effect, being partially sensitive to both EMC loss and Sec61 inhibition. Thus, EMC and Sec61 are largely non-redundant in terminal TMD insertion. The critical switchover point between EMC-dependence and Sec61-dependence is ~45 amino acids. This length is close to the translocation limit for the Oxa1 family (of which EMC is a member[46,47]) but long enough for the TMD to reach Sec61's lateral gate in the appropriate looped topology[18].

Charged residues in the C-terminal tail of 23L-SQS were seen to modestly but clearly reduce tail translocation (Fig. 4b). No difference was seen for tails with net positive or net negative charges. Importantly, all of these charge variants were translocated in an EMC-dependent manner. The charge-imperviousness for C-tail translocation of 23L-SQS was unexpected because EMC-mediated insertion of an N-terminal TMD (that is, a signal-anchor) or a tail-anchored protein is selectively disfavored if the tail to be translocated contains multiple positive charges[4,6,48]. The basis of this different behavior may relate to the length of time available for EMC-mediated insertion for a signal-anchor or tail-anchor versus the terminal TMD of a multipass protein. A ribosome displaying a signal-anchor has limited time for EMC-mediated insertion before docking onto Sec61, at which point EMC becomes inaccessible. Similarly, moderately hydrophobic tail-anchored proteins that are not promptly inserted by EMC can instead be inserted into mitochondria. Consistent with this interpretation, insertion of such signal-anchors and tail-anchors improves when Sec61 is depleted or when mitochondrial targeting is impaired, respectively[6,49]. By contrast, a multipass protein is already committed to endoplasmic reticulum insertion by the time EMC must insert the final tethered TMD, which would have prolonged and repeated access to EMC. Thus, EMC's substrate preference against positive charge translocation can be overcome by simply providing more time, implying that it can accommodate a broader range of substrates as a terminal TMD insertase.

A similar rationale probably explains the finding that in the context of 23L-SQS, EMC is able to accommodate a broad range of TMD hydrophobicity (Fig. 4c). For tail-anchored protein insertion, EMC-dependence is strongly influenced by hydrophobicity, with TMDs of higher hydrophobicity becoming dependent on the GET pathway. For example, SQS can progressively be made EMC-independent and GET-dependent by replacing less-hydrophobic residues in the TMD with leucine residues[14]. The same changes in 23L-SQS not only did not shift EMC-dependence but improved insertion overall. This indicates that EMC's preference for low-hydrophobicity TMDs seen for

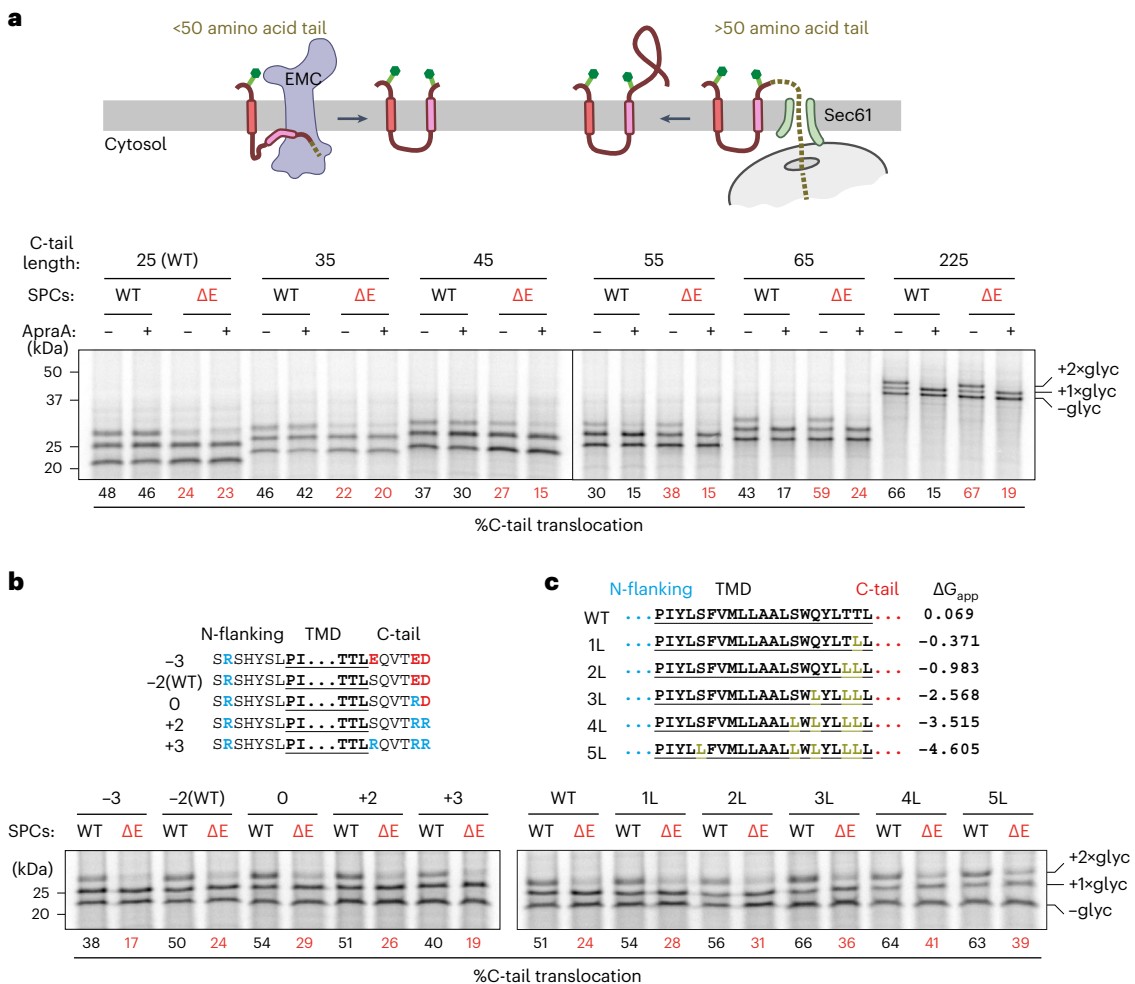

**Fig. 4 | Determinants of substrate C-tail translocation. a**, [35]S-methionine-labeled 23L-SQS variants with the different C-tail lengths indicated (top) translated in the presence of SPCs derived from WT or ΔE cells. Where indicated, 2 μM the Sec61 inhibitor ApraA was included in the translation reaction. Translation reactions were analyzed directly, and substrates with zero, one or two glycans attached are indicated. The two-glycan product is indicative of successful C-tail translocation. C-tail translocation was quantified by calculating the percentage of all glycosylated products that contain two glycans. **b**, [35]S-methionine-labeled 23L-SQS variants with charged residue mutations in the C-tail flanking the TMD were analyzed as in **a**. The sequences of the variants are shown with the net C-tail flanking charge indicated. **c**, [35]S-methionine-labeled 23L-SQS variants with leucine mutations in the TMD were analyzed as in **a**. The sequences of the variants are shown along with the hydrophobicity of each TMD as a $\Delta G_{app}$ score[65], in which negative values indicate favored membrane insertion.

tail-anchored proteins is not a reflection of EMC limitations but rather a consequence of high-hydrophobicity TMDs being captured by the GET pathway in the cytosol. Considered together, these findings illustrate that EMC's substrate specificity for terminal TMDs of multipass proteins is broad with respect to flanking charge and hydrophobicity but is limited by tail lengths shorter than ~50 amino acids.

**Immature membrane domains facilitate EMC targeting**
As noted above, 23L-GABRA1 shows EMC-dependent insertion of the terminal TMD (Extended Data Fig. 2) but with clearly lower efficiency than that seen in native GABRA1 (~38% compared to ~65%; Fig. 5). In considering possible explanations, we recognized that EMC has been proposed to function as a chaperone in addition to its insertase activity[33,41]. This suggested the possibility that in native GABRA1, TMD4 is brought in proximity to EMC by its preceding TMDs engaging EMC through its putative chaperone function. Replacing the first three TMDs of GABRA1 with 23L would eliminate this 'targeting' activity, explaining the loss of TMD4 insertion efficiency.

As a test of this idea, we asked whether TMD4 insertion could be rescued if TMD4 were preceded by an unrelated membrane domain that might be a putative chaperone substrate. Earlier studies have shown that until insertion of all of its TMDs, G-protein coupled receptors are targets for intramembrane chaperoning[50]. Therefore, we preceded TMD4 of GABRA1 with the first three TMDs of rhodopsin (termed Rho(1–3)) and tested the efficiency of terminal TMD insertion. To ensure that Rho(1–3) was not an EMC insertion substrate, we initiated its translocation using an N-terminal signal peptide and lumenal domain preceding its first TMD. Relative to 23L-GABRA1, terminal TMD insertion of Rho(1–3)-GABRA1 was markedly higher and similar to that seen with native GABRA1 (Fig. 5).

As Rho(1–3) is unrelated to TMD1–3 of GABRA1, these results indicate that the efficient insertion of TMD4 is not due to its complementarity with the earlier TMDs. Instead, the findings support a model in which the earlier TMDs target an otherwise poorly inserted TMD4 to EMC for efficient insertion. Consistent with this idea, the insertion of the SQS TMD improved when preceded by Rho(1–3) relative to 23L-SQS (Fig. 5), which itself was slightly more efficient than SQS inserted as a tail-anchored protein. This suggests that EMC-mediated insertion improves slightly by tethering a TMD close to the membrane (for example, with 23L) and improves further if

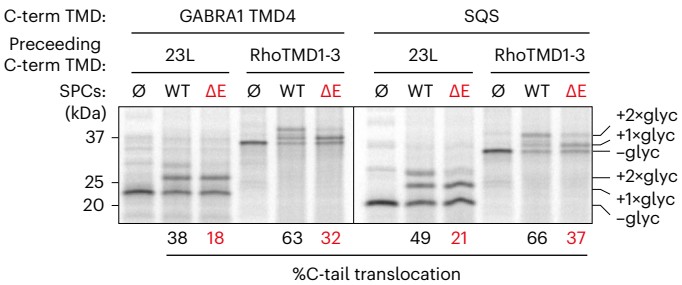

**Fig. 5 | Targeting to EMC facilitates C-terminal TMD insertion.** The GABRA1 TMD4 or SQS TMD was preceded by either the 23L TMD or TMD1–3 of rhodopsin, separated by a 100 amino acid linker. These constructs were translated in the absence (∅) or presence of SPCs from WT or ΔE cells and analyzed by SDS–PAGE and autoradiography. Substrates with zero, one or two glycans attached are labeled. Per cent C-tail translocation, calculated as in Fig. 4, is indicated.

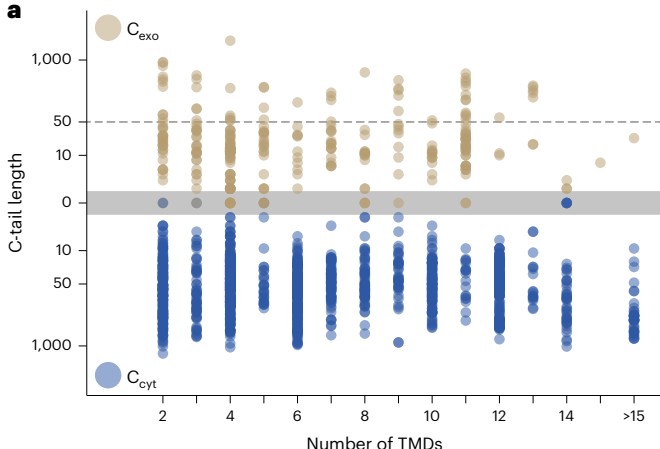

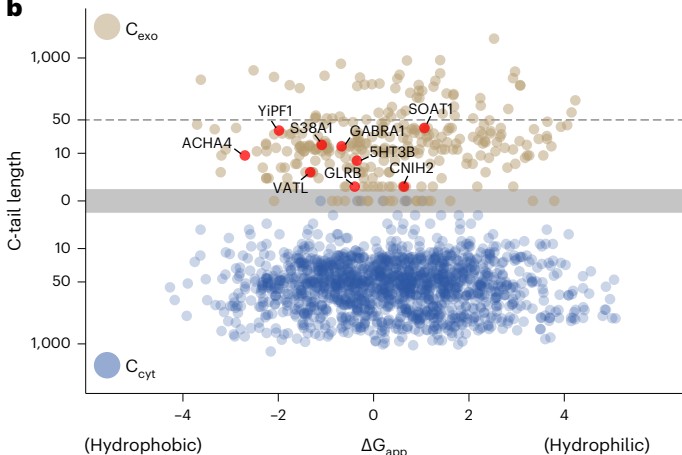

**Fig. 6 | Systematic analysis of multipass membrane protein orientation.**
**a**, The topology of all annotated multipass proteins in the human genome (1,784 total), determined using a combination of AlphaFold2 structure prediction to identify TMDs and the positive-inside rule to deduce orientation (see Supplementary Table 1). The C-tail locations ($C_{exo}$ or $C_{cyt}$ for exoplasmic and cytosolic, respectively) and lengths were tabulated and plotted versus the numbers of TMDs in the protein. **b**, The data from **a** were re-plotted relative to the $\Delta G_{app}$[65] score of the final TMD. Representative substrates analyzed in Fig. 7 for EMC-dependent translocation of their C-terminal TMDs are highlighted in red.

other parts of the protein are immature (for example, the first three TMDs of Rho or GABRA1).

## An expanded substrate repertoire of EMC

Using the insights gained from GABRA1 and 23L-SQS, we sought to predict all other terminal TMD substrates of EMC in the human genome. Leveraging the structural prediction afforded by AlphaFold2 (ref. 51) combined with the positive-inside rule[52], we manually curated the orientation of all 1,784 annotated multipass endoplasmic reticulum membrane proteins in the human genome. By plotting the C-tail lengths by topology, number of total TMDs and hydrophobicity of the terminal TMD (Fig. 6a,b), we found that 244 multipass proteins contain a terminal TMD whose downstream tail is non-cytosolic and 50 amino acids or shorter (Supplementary Table 1). From this manually curated list, we chose for analysis six proteins containing terminal TMDs of varying hydrophobicity and C-tail lengths. The functional expression of five of these proteins, or their close homologs, have been observed in earlier studies to be dependent on EMC[26,27,31,33]. Therefore, we tested their terminal TMDs in the 23L-TMD reporter with a C-terminal glycosylation site (Extended Data Fig. 4) to determine whether impaired C-tail translocation could explain their EMC dependence. Each of these showed at least partial dependence on EMC for C-terminal TMD insertion (Fig. 7a,b). The variable levels of insertion in the absence of EMC among the different substrates might reflect their capacity for unassisted insertion. Alternatively, they might be able to access other insertases such as GET, Sec61 or GEL depending on either TMD hydrophobicity or length of the C-tail, but this remains to be investigated.

We also tested two unrelated native proteins, SOAT1 and YIPF1, that also contain a terminal TMD with a translocated C-tail. These were chosen based on the prediction that earlier steps in their insertion would be EMC-independent, thereby allowing us to monitor terminal TMD insertion using a single C-tail glycosylation site (see Fig. 7a). As per our predictions (Fig. 6), both were observed to be partially EMC-dependent for C-tail translocation (Fig. 7c). As the separation of glycosylated from non-glycosylated products of full-length SOAT1 was challenging, we verified our conclusion with a better-resolved SOAT1 construct lacking the N-terminal cytosolic domain. EMC-dependent terminal TMD translocation of full-length and N-terminally deleted SOAT1 was further validated by selective recovery of the glycosylated products using the lectin conconavalin A. Of note, both endogenous and exogenously expressed SOAT1 were shown in earlier studies to be strongly dependent on EMC in cells, but the basis of this dependence was not clear[32]. Our finding that insertion of the final TMD of SOAT1 is EMC-dependent to a comparable level as SOAT1 biogenesis in cells now provides an explanation. The ~250 new putative substrates of EMC are highly diverse in their topology and function and may help to explain

the complex and pleiotropic phenotypes of EMC loss in a wide range of organisms[7,53].

## Discussion

Our study provides three insights into the problem of membrane protein topogenesis. First, we reveal that unlike previous suggestions[4,14,33], EMC is not specific for poorly hydrophilic TMDs and does not have a strong discriminatory capacity against the translocation of positive charges. Instead, it seems that EMC's preferences against positive charges and high hydrophobicity that were observed in earlier studies are consequences of competing reactions. Thus, EMC's intrinsic capacity for TMD insertion is broader than had been thought. Second, various substrates that show an EMC requirement in cells, such as pentameric ion channels[26,27] and SOAT1[32], can probably be ascribed to a failure of terminal TMD insertion. These substrates had been puzzling because they are neither tail-anchored proteins nor initiated with a signal-anchor in the $N_{exo}$ topology (in which the N terminus faces the exoplasmic side of the membrane), which previously were the known targets for EMC function. Although we cannot exclude additional roles for EMC

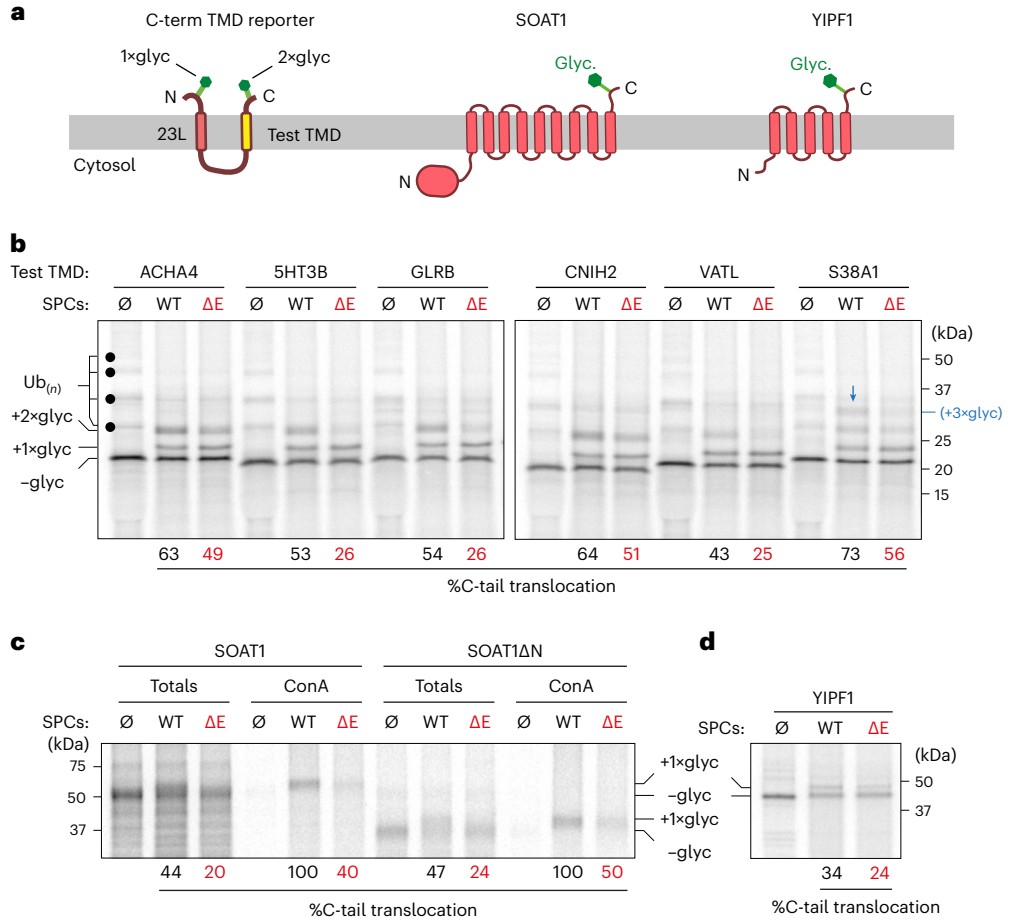

**Fig. 7 | EMC facilitates C-terminal TMD insertion of distinct classes of substrates. a**, Domain organization of a C-terminal TMD reporter, which contains 23L and test TMD (yellow), a cytosolic loop (~100 amino acids) and two glycosylation sites in both N-tail and C-tail (left); topology and domains of full-length SOAT1 (middle) and YIPF1 (right) containing C-terminal glycosylation sites. **b**, [35]S-methionine-labeled C-terminal TMD reporters with indicated TMDs, translated in the absence (Ø) or presence of SPCs from WT or ΔE cells. Translation reactions were analyzed directly by SDS–PAGE and autoradiography. The positions of translated reporters with zero, one or two glycans are indicated. In the absence of endoplasmic reticulum membranes, cytosolic quality control factors recognize non-translocated substrates and modify them with mono- or poly-ubiquitin (Ub$_{(n)}$) as indicated by the black dots. Note that S38A1 contains two C-tail glycosylation sites, resulting in a triply glycosylated product (blue arrow). C-tail translocation was quantified as previous figures. **c**, [35]S-methionine-labeled SOAT1 or an N-terminal domain deletion of SOAT1 (SOAT1ΔN), translated in the absence (Ø) or presence of SPCs from WT or ΔE cells. One aliquot was subject to denaturing immunoprecipitation through an N-terminal HA tag (Totals); another was subjected to conconavalin A pulldown (ConA) to recover the glycosylated population. Non-glycosylated and glycosylated products are indicated, and percent glycosylation was calculated. In ConA recovered lanes, glycosylation was normalized to WT. **d**, [35]S-methionine-labeled YIPF1 was translated in the absence (Ø) or presence of SPCs from WT or ΔE cells and analyzed directly by SDS–PAGE and autoradiography. Substrate glycosylation state and per cent C-tail translocation are indicated.

in GABRA1 or SOAT1, the magnitude of the defect seen in C-terminal TMD insertion can explain the consequence in cells.

Third, and perhaps of most conceptual importance, is the finding that the topology of multipass proteins can be determined by a combination of co-translational and post-translational reactions. Co-translational events result in committing the protein for endoplasmic reticulum targeting and insertion of most TMDs, whereas other TMDs can be inserted post-translationally after the substrate has presumably departed the ribosome–translocon complex. Diffusion away from the ribosome (or dissociation of the ribosome from the membrane) would be a prerequisite for EMC-mediated insertion because EMC cannot access substrates near the exit tunnel of a Sec61-bound ribosome[6,22]. Hence, our findings show that at least some multipass proteins are released from the ribosome–translocon complex in a topologically immature form, with EMC rectifying the topology at a later step. Targeting to EMC for this post-translational rectification step may be facilitated by EMC's putative chaperone activity[33,41], which would preferentially engage immature membrane-embedded

domains such as those lacking a terminal TMD. In cells, a failure to rectify the topology promptly would presumably lead to engagement by quality control factors within the membrane of the topologically incomplete protein[54,55] or in the cytosol of the uninserted TMD[45,56,57], thereby explaining why these EMC substrates are degraded in ΔEMC cells.

Although we have demonstrated post-translational topological rectification for C-terminal TMDs, it is plausible that in other circumstances, pairs of TMDs separated by a short loop are similarly inserted post-translationally (for example, Fig. 1b). The Oxa1 family (which, in eukaryotes, is composed of EMC, GEL and GET complexes) has been shown to be capable of such insertion[8,10,58–60], which might be needed if some TMDs are skipped during co-translational insertion. Skipping of TMDs followed by post-translational rearrangement has been suggested in earlier work[61], but the basis of such a mechanism is unclear. Our findings now suggest a plausible mechanism utilizing an Oxa1 family member, which is an idea that warrants experimental analysis in future work.

Our study adds to the emerging principle that there is a straightforward segregation of function between the Oxa1 family and the SecY family[3,9]. Oxa1 family members mediate TMD insertion when the translocated segment of a flanking polypeptide is shorter than ~50 amino acids, whereas Sec61 (or SecY in prokaryotes) mediates TMD insertion when the translocated flanking domain is longer. In eukaryotes, tail-anchored proteins are inserted by the GET and EMC insertases[14,62,63], signal-anchored proteins with a short translocated N-terminal tail are inserted by EMC[4,6], internal TMD pairs with short loops are inserted by GEL[8,10] and, as shown here, terminal TMDs with a short translocated C-tail use EMC. In bacteria, the sole Oxa1 family member YidC would perform all of these jobs, perhaps explaining why it is essential[64]. The larger and more diverse membrane proteome in eukaryotes might have driven an expansion and specialization of endoplasmic-reticulum-localized Oxa1 family members, which perhaps also affords a degree of redundancy and robustness to the essential process of membrane protein biogenesis.

## Online content

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

## Methods

### Cell culture and cell lines

All cells were cultured at 37°C with 5% $CO_2$ in DMEM (Gibco, 10569-010) supplemented with 10% fetal bovine serum (Gibco, 10270106). Wild-type and ΔEMC6 HEK293 cells have been previously described[14]. The ΔTMCO1 HEK293 cell line was obtained from R. Keenan[20]. ΔEMC6ΔTMCO1 double-knockout cells were generated by knocking out TMCO1 in ΔEMC6 cells. Ribonucleoprotein complexes were formed between Alt-R S.p. Cas9-GFP V3 (IDT, 10008100) and Alt-R CRISPR-Cas9 sgRNA (5′-ACTTGTCTGTCCTGTAAACC-3′; IDT) following the manufacturer's recommendations. Ribonucleoprotein complexes were transfected into ΔEMC6 cells using Lipofectamine RNAiMAX (Invitrogen, 13778150) according to the manufacturer's protocol; 48 h later, green fluorescent protein (GFP)-positive cells were sorted into single colonies and expanded. Knockout cells were screened by immunoblotting. Flp-In T-REx 293 cells stably expressing wild-type EMC3-FLAG (NP_060917.1) or EMC3-FLAG variants (M$^{cyt-1}$-S: M101, 106, 110, 111S; R31A; F148L; R13E) were generated by integrating each construct into the FRT site and selecting for Flp-mediated recombination through 100 µg ml$^{-1}$ hygromycin B for 2 weeks[6]. The tetracycline-inducible 293 cell line expressing the human $GABA_A$ receptor has been previously described[29]. In this cell line, the α1 subunit (NP_001178048.1) in the pcDNA4-TO-Zeocin backbone is FLAG-tagged after its 27 amino acid signal sequence; the β3 subunit (NP_068712.1) in a pcDNA3.1-TO-Hygromycin backbone is untagged; the γ2L subunit (NP_944494.1) in a pACMV-TO-blasticidin backbone is 1D4-tagged (TETSQVAPA) at the C terminus after a (GGS)$_3$GK linker. Tetracycline-inducible 293 cell lines expressing GFP-P2A-RFP-SQS (NP_004453.3, aa378–410) and GFP-P2A-RFP-ASGR1 (NP_001662.1) reporters in the pcDNA5-FRT-TO backbone were generated by stably integrating reporter plasmids into the FRT site and have been previously described[4].

### Recombinant DNA reagents

Plasmids or gBlocks (IDT) used for in vitro transcription and translation assays contained an SP6 promoter and coding sequences. All plasmids were verified by sequencing. Wild-type GABRA1 (NP_001345964.1) contains two mutations (V436M and L448M) to facilitate the detection of the C-terminal domain by autoradiography without changing the TMD length, hydrophobicity or C-tail charge. GABRA1-glyc was generated by adding an opsin tag (MNGTEGPNFYVPFSNKTVD)[66] to the C terminus of wild-type GABRA1. SQS-glyc (NP_004453.3, aa378–410) has been previously described[14]. The 23L-GABRA1 contains an N-terminal 9×His tag, a glycosylation sequence, a soluble tail from β1-adrenergic receptor (NP_001290104.1, residues 29–44), 23 leucine codons, a soluble cytosolic loop (GGSG-mEGFP(1–92)), TMD4 and flanking regions of GABRA1 (NP_001345964.1, aa 407–455) and an opsin tag. 23L-SQS replaces TMD4 and flanking sequences of 23L-GABRA1 with the TMD and flanking regions of SQS (NP_004453.3, aa378–410). The following 23L-SQS variants, used in Figs. 3 and 4, were made by site-directed mutagenesis: P202C (cysteine in C-tail for EMC crosslinking); P202Amber (for incorporation of a photoreactive amino acid into the C-tail); S168C (cysteine in TMD); S185E (−3); E189R (0); E189R,D190R (+2); S185R,E189R,D190R (+3); T183L (1L); T182L,T183L (2L); Q179L,T182L,T183L (3L); S177L,Q179L,T182L,T183L (4L); and S168L,S177L,Q179L,T182L,T183L (5L). Extensions of the C-tail length to a total length of 25, 35, 45, 55, 65 and 225 amino acids were generated by inserting part of the coding sequence for mCherry. The Rho(1–3) domain that precedes the cytosolic loop in the GABRA1 and SQS constructs shown in Fig. 5 contains the following: the prolactin signal sequence (NP_776378, aa1–33), a Twin-Strep-tag (SAWSHPQFEKGGGSGGGSGGGSAWSHPQFEK), a linker (AGGSAGSGGGSAGGSA), the VHP domain (NP_990773, aa792–826), a glycosylation site, a linker (GGGSAGGGSA) and rhodopsin (NP_001014890, aa32–152). For the reporter constructs shown in Fig. 7b, TMD4 and flanking

regions of 23L-GABRA1 were replaced by the following sequences: CNIH2 (NP_872359, aa126–160); VATL (NP_001685, aa119–155); S38A1 (NP_109599, aa440–487); ACHA4 (NP_000735, aa588–627); 5HT3B (NP_006019, aa402–441); and GLRB (NP_001159532, aa463–497). SOAT1 (NP_003092.4) was appended at the C terminus with the opsin glycosylation tag. The N-terminal deletion of SOAT1 removed amino acids 2–125. YIPF1 (NP_061855.1) was obtained from R. Keenan.

### Small interfering RNA knockdown and flow cytometry

For monitoring the surface expression of $GABA_A$ receptors, either negative control small interfering RNA (siRNA) (Invitrogen, 4390843) or EMC4 siRNA (Ambion, s27733) was transfected into the 293 cell line expressing the human $GABA_A$ receptor using Lipofectamine RNAiMAX according to the manufacturer's protocol. A final concentration of 10 nM of siRNA was used and the total knockdown time was 66 h. At 60 h, doxycycline was added to a final concentration of 0.1 µg ml$^{-1}$ to induce expression of the $GABA_A$ receptor for 6 h. Cells were then collected and subjected to surface labeling. Pelleted cells were resuspended with 100 µl of cold PBS, supplemented with 1 µl of phycoerythrin-labeled FLAG antibody (BioLegend, 637310) and incubated in the dark at 4°C for 1 h. Cells were then washed with cold PBS, passed through a 70 µm filter and then analyzed on a BD LSR II flow cytometer for appropriate fluorescent channels. A total of 30,000 events were analyzed, and phycoerythrin fluorescence, reflective of $GABA_A$ receptor surface levels, was plotted as a histogram using FlowJo. To analyze the stability of ASGR1 or SQS, 293 cell lines stably expressing GFP-P2A-RFP-ASGR1 or GFP-P2A-RFP-SQS were used as previously described[50]. The P2A sequence in these constructs causes ribosome skipping, resulting in the translation of equimolar amounts of GFP and the RFP-tagged protein. Therefore, a steady-state RFP:GFP ratio reflects the stability of the RFP-tagged protein. Failure in biogenesis will lead to degradation by cellular quality control pathways and a decreased RFP:GFP ratio. Knockdown was performed as for $GABA_A$ receptor cells, with the GFP and RFP fluorescence monitored by flow cytometry on 30,000 cells. The RFP:GFP ratio was plotted as a histogram using FlowJo.

### Preparation of SPCs

Cells at 95–100% confluency were trypsinized and collected by centrifugation at 4°C, washed once with ice-cold 1×PBS and resuspended in 1×RNC buffer (50 mM HEPES, pH 7.4, 100 mM KOAc, 5 mM Mg(OAc)$_2$) containing 0.01% purified digitonin[67]. SPCs were pelleted and washed once with 1×RNC. To digest endogenous mRNAs, SPCs were resuspended in 100 µl of 1×RNC containing 1 mM CaCl$_2$ and 150 units per ml micrococcal nuclease (Roche, 10107921001). Nuclease digestion was performed for 10 min at room temperature (20°C) and was terminated by adding a final concentration of 2 mM EGTA. Nuclease-digested SPCs were pelleted, washed once with 1×RNC buffer, resuspended in 0.5×RNC buffer to 6,000–10,000 cells per ml and used immediately in translocation assays.

### Preparation of endoplasmic-reticulum-enriched membranes

Approximately 80% of confluence cells were collected by trypsinization. Cells were pelleted, washed once with cold 1×PBS and flash-frozen in liquid nitrogen. Thawed cell pellets were mixed with 5 volumes of 20 mM HEPES, pH 7.4; 5 mM KCl; 1.5 mM MgCl$_2$; 2 mM dithiothreitol (DTT); and protease inhibitor (Roche, 10106399001) and incubated on ice for 15 min. Cells were lysed on ice by 35 strokes of dounce homogenization (DWK Life Sciences, 357542). Cell lysates were adjusted to 20 mM HEPES, pH 7.4; 210 mM mannitol; 70 mM sucrose; 0.5 mM EDTA; 2 mM DTT; and protease inhibitor. Cell debris and nuclei were cleared by centrifugation at 4°C for 10 min at 700×*g*. Membranes were then pelleted by centrifugation at 4°C for 10 min at 8,500×*g*, washed once and resuspended in a buffer containing 20 mM HEPES, pH 7.4; 210 mM mannitol; 70 mM sucrose; 0.5 mM EDTA; 2 mM DTT; and protease inhibitor to give an OD280 of 20. Different amounts of resuspended

membranes were used to assay the levels of key translocation components by blotting.

## In vitro transcription and translation

Transcription reactions with SP6 polymerase were performed at 37°C for 1 h and contained the following components: DNA that encodes regions of interest for translation reactions (PCR-amplified and purified by Qiagen PCR purification kit, 10 ng μl$^{-1}$); HEPES, pH 7.4 (40 mM); spermidine (2 mM; Sigma, S0266); RNA cap structure analog (0.33 mM; NEB, S1404L); reduced glutathione (10 mM); MgCl$_2$ (6 mM); NTPs (0.5 mM each for ATP; Roche, 10519979001), CTP (Sigma, C1506) and UTP (Sigma, U6875), 0.1 mM for GTP (Roche, 10106399001); SP6 RNA polymerase (0.4 U μl$^{-1}$; NEB, M0207L); and RNase inhibitor (0.8 U μl$^{-1}$; Promega, N2515).

Translation reactions were performed at 32°C for 30 min and contained the following components: micrococcal nuclease-digested rabbit reticulocyte lysates (Green Hectares) (34% of the total volume); transcription reaction from the previous step (5% volume); SPCs (10% volume); ATP and GTP (1 mM each); an ATP regeneration system (creatine phosphate (12 mM; Roche, 10621714001); creatine kinase (0.04 mg ml$^{-1}$; Roche, 10127566001)); spermidine (0.3 mM); HEPES, pH 7.4 (20 mM); KOAc (50 mM); Mg(OAc)$_2$ (2 mM); reduced glutathione (1 mM); tRNAs purified from pig liver (0.05 mg ml$^{-1}$); 19 of the 20 amino acid except for methionine (40 μM each; Promega, L9961); and $^{35}$S-methionine (0.5 μCi μl$^{-1}$; PerkinElmer, NEG709A001MC).

For incorporating the photoreactive amino acid p-benzoyl-l-phenylalanine (Bpa) through amber suppression, the following components are included in the translation reaction[68]: suppressor *Bacillus stearothermophilus* tRNA$_{CUA}$$^{Tyr}$ (5 μM); *E. coli* Bpa tRNA synthetase (0.25 μM); and Bpa (100 μM). These components were pre-mixed into a 10× solution (in 50 mM HEPES, pH 7.4, 100 mM KOAc, 1 mM Mg(OAC)$_2$) and were pre-incubated at 32°C for 15 min before adding to the translation reaction. Where indicated, the Sec61 lateral gate inhibitor ApraA (obtained from V. Paavilainen and K. McPhail) was included in the translation reaction at 2 μM.

## Protease protection assays

The 60 μl in vitro translation reactions were chilled on ice, and the SPCs were pelleted (20,000×*g* for 2 min), washed once with 1×RNC buffer (50 mM HEPES, pH 7.4, 100 mM KOAc, 5 mM Mg(OAc)$_2$) and resuspended in 30 μl 0.5×RNC buffer. Samples lacking SPCs were used directly without pelleting. Samples were divided into two aliquots; one aliquot (two-thirds of the total volume) was adjusted to 0.5 mg ml$^{-1}$ proteinase K and incubated on ice for 50 min. Proteinase K was quenched by adding 250 mM of PMSF for 2 min, then transferring the entire reaction to a tenfold excess volume of 1% SDS, 100 mM Tris-HCl, pH 8.0 pre-heated to 100°C and heated for 10 min. The samples were either analyzed directly or subjected to immunoprecipitation as indicated in the figure legends.

## Site-specific crosslinking

The 120 μl in vitro translation reactions were used for bismaleimidohexane (BMH) crosslinking experiments. All steps following the translation reaction were at 4°C until the reaction was denatured in SDS. SPCs were pelleted and resuspended in 60 μl 0.5×RNC buffer. One aliquot was removed as the no-crosslinking control and the remainder of the sample was adjusted to 250 μM BMH (Thermo Scientific, 22330) and incubated on ice for 10 min. The crosslinking reaction was quenched by adjusting the final concentration of DTT to 25 mM. After denaturation in 1% SDS, 100 mM Tris-HCl, pH 8.0, the samples were either analyzed directly or processed further for immunoprecipitation or deglycosylation. SMPH (Succinimidyl 6-((beta-maleimidopropionamido) hexanoate)) and UV crosslinking experiments were performed similarly to the BMH crosslinking experiments, with the following differences: SMPH (Thermo Scientific, 22363) crosslinking (200 μl total reaction)

was at 200 μM final concentration for 30 min, and quenched with 50 mM Tris-HCl pH 7.4 and 5 mM DTT; UV crosslinking (100 μl total reaction) was on ice with UV irradiation by a UVP Blak-Ray B-100AP high-intensity lamp with the bulb positioned ~10 cm above the samples.

## Immunoprecipitation and PNGase F treatment

Denaturing immunoprecipitation after crosslinking, proteinase K digestion and glycanase digestion have been previously described[6]. SDS-denatured samples were diluted tenfold in ice-cold immuno-precipitation buffer (1×PBS, 250 mM NaCl, 0.5% TX-100, 10 mM imi-dazole) and mixed with 2.5 μl of anti-FLAG resin (Millipore, A2220), 2.5 μl of Monoclonal Anti-HA resin (Millipore, A2095), or 5 μl of protein A resin (Repligen, CA-HF-0100) along with the appropriate antibody. A total of 1.25 μg of GABRA1 antibody (Invitrogen, PA5-79291) was used per immunoprecipitation. The mixture was rotated end-over-end for 1.5 h (for FLAG or HA) or 3 h (for GABRA1 immunoprecipitation) at 4°C. Beads were washed twice with cold immunoprecipitation buffer and eluted by boiling in 10 μl of 2.5× SDS−PAGE sample buffer for 10 min. For deglycosylation experiments, crosslinked samples (Fig. 3) or total translation reactions (Extended Data Fig. 4) were split into two halves after denaturation with 0.5% SDS and 50 mM Tris-HCl, pH 8. One half was untreated and the other was adjusted to 1% NP-40, 1× GlycoBuffer 2 and 25 U ml$^{-1}$ of PNGase F (NEB, P0704S) and digested at 32°C for 30 min. Both halves were subjected to immunoprecipitation as described above (Fig. 3) or analyzed directly (Extended Data Fig. 4).

## Bioinformatic analysis of membrane proteins

All proteins containing TMDs were retrieved from the UniProt data-base[69]. The UniProt annotations were used to define the start and end of the TMD helices. Proteins containing a single TMD or multipass membrane proteins localized to mitochondria were manually removed from this set. The AlphaFold2 (ref. 51) predicted structure, available from the UniProt database for each of the remaining 1,784 multipass membrane proteins, was inspected manually to annotate the number of TMD helices and the overall charge of each TMD-flanking side of the structure. The overall basic flank was designated cytosolic as per the positive-inside rule[52]. Then the C-terminal TMD was identified and assigned the appropriate orientation, hydrophobicity (as calculated using the $\Delta G_{app}$ predictor[65]) and flanking C-tail length. C-tails facing the cytosol were designated 'C$_{cyt}$' and those facing the opposite orientation were designated 'C$_{exo}$'. The curated list is provided in Supplementary Table 1; information from this table was used to generate the plots in Fig. 6.

## SDS−PAGE and blotting

Cell lysates or endoplasmic-reticulum-enriched membranes were analyzed by SDS−PAGE on 12% Tris-Tricine gels. SDS−PAGE gels were transferred to a nitrocellulose membrane (Biorad, 1620112) and blotting was performed with standard procedures using 5% non-fat dried milk as the blocking agent. The following antibodies and dilutions were used for blotting: CCDC47 (Bethyl Laboratories, A305-100A; 1:5,000); EMC3 (Invitrogen, 711771; 1:5,000); EMC6 (Abcam, ab84902; 1:1,000); Calnexin (Enzo, ADI-SPA-865; 1:5,000); Sec61α (ref. 70; 1:5,000); TMCO1 (Invitrogen, PA5-43350; 1:500); Sec61β (ref. 71; 1:5,000); CAML (Cell Signaling Technology, 13913S; 1:1,000); FLAG M2-HRP (Sigma, A8592; 1:5,000); EMC4 (Abcam, ab123719; 1:2,000); and β-Actin-HRP (Sigma, A3854; 1:10,000).

## Quantification of C-tail translocation

Quantification was performed on raw phosphorimager files using Fiji. The pixel intensity and area of each band were measured, from which the background intensity was subtracted. For the C-terminal TMD reporters, C-tail translocation was calculated by dividing the value for the C-tail translocated band (typically the 2×-glycosylated

product) by the sum of total membrane inserted bands (typically the 1×-glycosylated and 2×-glycosylated bands). In the case of SQS-glyc (Fig. 1d,e), SOAT1 (Fig. 7c) and YIPF1 (Fig. 7d), per cent C-tail translocation is calculated by dividing the intensity of the glycosylated band by the sum of glycosylated and non-glycosylated bands.

## Statistics and reproducibility

This study does not contain any statistical analysis. All data presented in this paper have been reproduced in independent experiments. The number of independent experiments are indicated in parentheses for the following main and extended data figure panels: Fig. 2a (2); Fig. 2b (2); Fig. 2c (2); Fig. 2d (3); Fig. 3a (3); Fig. 3b (3); Fig. 3c (2); Fig. 3d (3); Fig. 4a (2); Fig. 4b (2); Fig. 4c (2); Fig. 5 (2); Fig. 7b (2); Fig. 7c (4 for SOAT1, 2 for SOAT1ΔN); Fig. 7d (6); Extended Data Fig. 1 (2); Extended Data Fig. 2 (2); Extended Data Fig. 3 (2); Extended Data Fig. 4 (2).

## Reporting summary

Further information on research design is available in the Nature Portfolio Reporting Summary linked to this article.

## Data availability

All data are provided in the Main, Extended Data and Supplementary Information. Publicly available data sets used in this study were obtained from the UniProt database (http://www.uniprot.org) and the AlphaFold2 protein structure database (https://www.alphafold.ebi.ac.uk). Source data are provided with this paper.

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

## Acknowledgements

We thank H. Wang and other Hegde Lab members for fruitful discussions, Y. Hooda for flow cytometry data, R. Judy for help with data curation, S. Juszkiewicz for plasmids, V. Paavilainen and K. McPhail for providing Apratoxin A and R. Keenan for discussions and reagents. This work was supported by the UK Medical Research Council (grant MC_UP_A022_1007 to RSH) and a European Molecular Biology Organization (EMBO) Postdoctoral Fellowship (ALTF 369-2021 to H.W.).

## Author contributions

H.W. performed all experiments in the study. L.S. generated the ΔEMC6ΔTMCO1 cell line. H.W. and R.S.H. conceived the project and R.S.H. provided guidance and mentoring. H.W. and R.S.H. wrote the paper.

## Competing interests

The authors declare no competing interests.

## Additional information

**Extended data** is available for this paper at https://doi.org/10.1038/s41594-023-01120-6.

**Correspondence and requests for materials** should be addressed to Ramanujan S. Hegde.

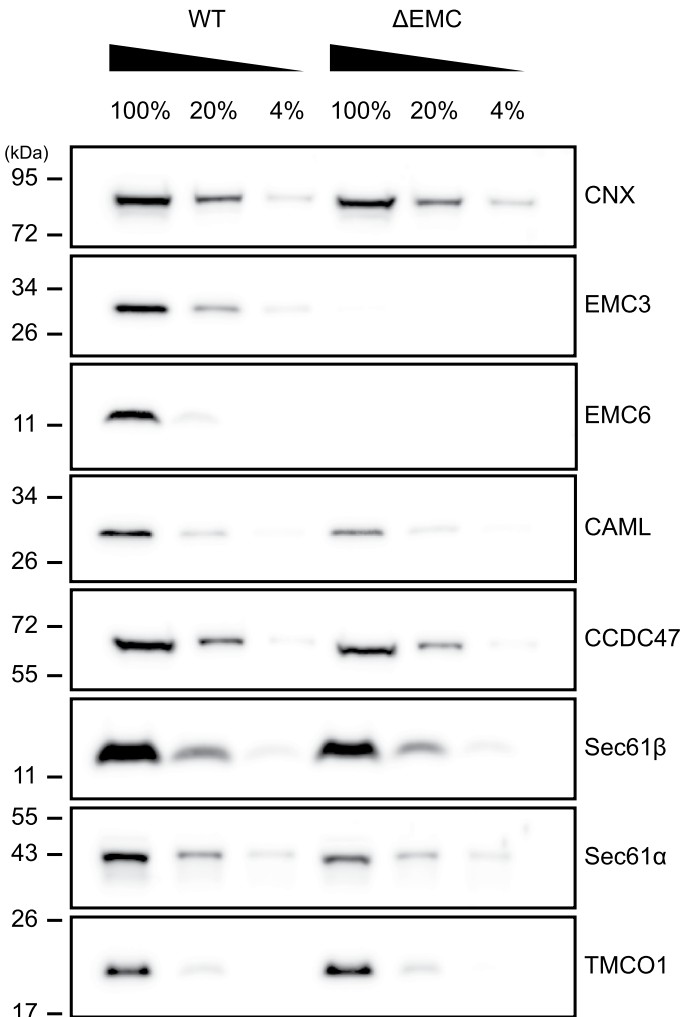

**Extended Data Fig. 1 | Characterization of ΔEMC cells.** Serial dilutions of total microsomes from WT or ΔEMC cells were analyzed by SDS-PAGE and immunoblotting for the indicated ER-resident proteins. These data complement earlier analysis for additional ER proteins (including Sec62, Sec63, TRAM, TRAPα, the 12 kD subunit of the signal peptidase complex, and other EMC subunits) using the same parental and ΔEMC cells[4].

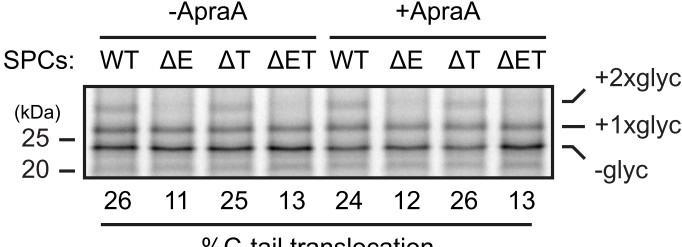

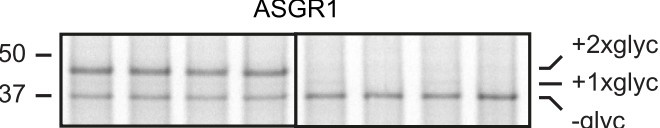

**Extended Data Fig. 2 | Characterization of GABRA1 C-terminal TMD insertion.** Domain diagram of 23L-GABRA1 is shown (top): a glycosylated N-tail is followed by a TMD comprised of 23 leucine residues (23L), a cytosolic loop (~100aa), and TMD4 of GABRA1 that is followed by a translocated and glycosylated C-tail. [35]S-methionine labeled 23L-GABRA1 (middle) or ASGR1 (bottom) were translated in the presence of SPCs from WT, ΔEMC (ΔE), ΔTMCO1 (ΔT) or ΔEMCΔTMCO1 (ΔET) cells. Translated products were analyzed directly by SDS-PAGE and autoradiography. Where indicated, Sec61 inhibitor ApraA was included in the translation reaction. Substrates with different glycosylation states are indicated. C-tail translocation of 23L-GABRA1 was quantified by calculating the percentage of double glycosylated products among all glycosylated products.

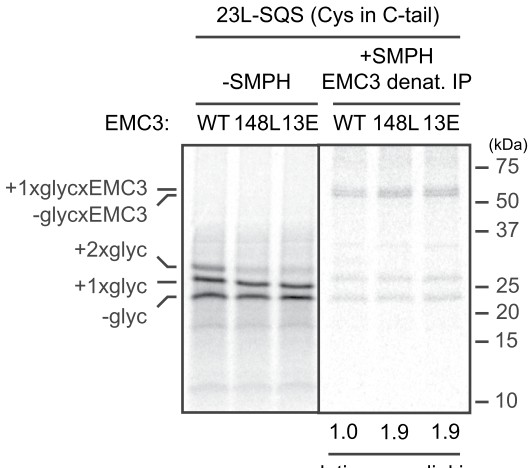

**Extended Data Fig. 3 | Analysis of EMC3-substrate crosslinking in EMC3 mutants.** [35]S-methionine labeled 23L-SQS that contains a single cysteine within its C-tail was translated in the presence of SPCs derived from cells expressing FLAG tagged WT or each of two EMC3 mutants that impair its insertase function (F148L or R13E). Translation products were analyzed directly (-SMPH) or subject to SMPH crosslinking and FLAG denaturing IP (+SMPH, EMC3 denat. IP). Differentially glycosylated substrates and EMC3 crosslinks are indicated. Relative crosslinking was quantified by normalizing intensity of crosslinked products relative to totals. Note that earlier analysis has shown that these mutants do not affect either the expression level or assembly of EMC[6].

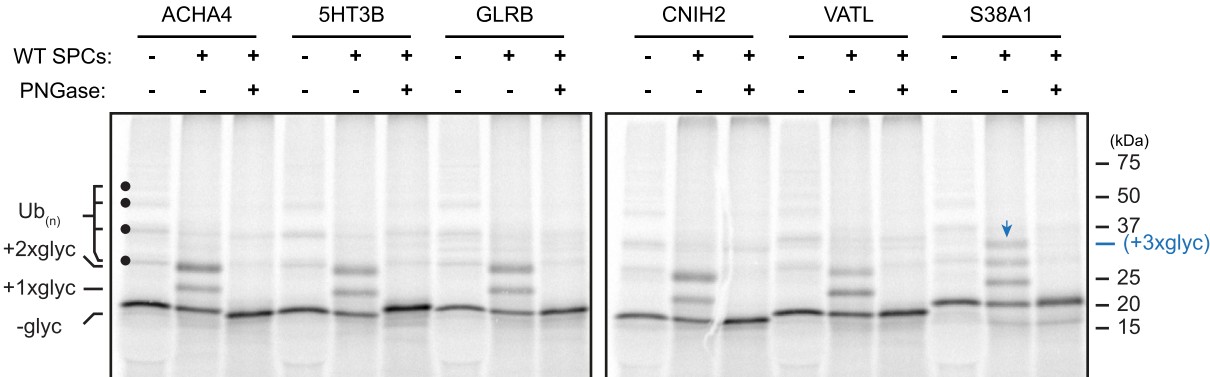

**Extended Data Fig. 4 | Glycosylation analysis of C-terminal TMD reporters.** The diagram shows the domain organization of the C-terminal TMD reporters consisting of an N-terminal translocated tail, a TMD with 23 leucines, a cytosolic loop (~100aa) and a test TMD followed by a C-terminal translocated tail. If properly inserted, both N- and C-tails are glycosylated. $^{35}$S-methionine labeled reporters were translated in the absence or presence of WT SPCs. Translated products were either analyzed directly or subject to PNGase F digestion to definitively identify the glycosylated products as indicated. Ubiquitinated non-translocated products [$Ub_{(n)}$] are indicated in black dots. A triply glycosylated form of S38A1 due to a second consensus site in the C-tail is marked by a blue arrow.

# Reporting Summary

## Statistics

For all statistical analyses, confirm that the following items are present in the figure legend, table legend, main text, or Methods section.

| n/a | Confirmed | |
|---|---|---|
| ☐ | ☒ | The exact sample size (*n*) for each experimental group/condition, given as a discrete number and unit of measurement |
| ☐ | ☒ | A statement on whether measurements were taken from distinct samples or whether the same sample was measured repeatedly |
| ☒ | ☐ | The statistical test(s) used AND whether they are one- or two-sided<br>*Only common tests should be described solely by name; describe more complex techniques in the Methods section.* |
| ☒ | ☐ | A description of all covariates tested |
| ☒ | ☐ | A description of any assumptions or corrections, such as tests of normality and adjustment for multiple comparisons |
| ☐ | ☒ | A full description of the statistical parameters including central tendency (e.g. means) or other basic estimates (e.g. regression coefficient) AND variation (e.g. standard deviation) or associated estimates of uncertainty (e.g. confidence intervals) |
| ☒ | ☐ | For null hypothesis testing, the test statistic (e.g. *F*, *t*, *r*) with confidence intervals, effect sizes, degrees of freedom and *P* value noted<br>*Give P values as exact values whenever suitable.* |
| ☒ | ☐ | For Bayesian analysis, information on the choice of priors and Markov chain Monte Carlo settings |
| ☒ | ☐ | For hierarchical and complex designs, identification of the appropriate level for tests and full reporting of outcomes |
| ☒ | ☐ | Estimates of effect sizes (e.g. Cohen's *d*, Pearson's *r*), indicating how they were calculated |

*Our web collection on statistics for biologists contains articles on many of the points above.*

## Software and code

Policy information about availability of computer code

| | |
|---|---|
| Data collection | Phosphorimaging of radioactive samples was acquired on Typhoon FLA7000 (GE Healthcare). Flow cytometry data were collected using a Beckton Dickinson LSRII. |
| Data analysis | Fiji (Version 1.53c) was used to analyze band intensity in autoradiography. FlowJo (version 10.8.0) was used to analyze flow cytometry data. |

For manuscripts utilizing custom algorithms or software that are central to the research but not yet described in published literature, software must be made available to editors and reviewers. We strongly encourage code deposition in a community repository (e.g. GitHub). See the Nature Portfolio guidelines for submitting code & software for further information.

## Data

Policy information about availability of data

All manuscripts must include a data availability statement. This statement should provide the following information, where applicable:
- Accession codes, unique identifiers, or web links for publicly available datasets
- A description of any restrictions on data availability
- For clinical datasets or third party data, please ensure that the statement adheres to our policy

The paper contains all the data that was used to arrive at the conclusions of this study in main, extended and supplemental items. Publicly available datasets used in this study: Uniprot database (http://www.uniprot.org/), AlphaFold2 protein structure database (https://www.alphafold.ebi.ac.uk/).

# Research involving human participants, their data, or biological material

Policy information about studies with [human participants or human data](). See also policy information about [sex, gender (identity/presentation), and sexual orientation]() and [race, ethnicity and racism]().

| | |
|---|---|
| Reporting on sex and gender | NA |
| Reporting on race, ethnicity, or other socially relevant groupings | NA |
| Population characteristics | NA |
| Recruitment | NA |
| Ethics oversight | NA |

Note that full information on the approval of the study protocol must also be provided in the manuscript.

# Field-specific reporting

Please select the one below that is the best fit for your research. If you are not sure, read the appropriate sections before making your selection.

☒ Life sciences        ☐ Behavioural & social sciences        ☐ Ecological, evolutionary & environmental sciences

For a reference copy of the document with all sections, see [nature.com/documents/nr-reporting-summary-flat.pdf]()

# Life sciences study design

All studies must disclose on these points even when the disclosure is negative.

| | |
|---|---|
| Sample size | No sample size calculations were performed. Biochemical experiments and flow cytometry experiments were repeated on independent days to verify reproducibility. Flow cytometry measurements included a minimum of 30,000 live cells expressing the reporter. Each experiment was performed at least twice to verify that the same result was obtained in each case. The specific number of repeats for every experiment is included in the Methods section. The minimum sample size of two was chosen because extensive earlier published data on these types of bulk biochemical assays show that the type of processes we are studying displays very little variability from experiment to experiment. The number of cells chosen for analysis in flow cytometry was also based on prior published studies on this type of assay demonstrating that minimal variability is seen when 10,000 or more cells are analyzed. |
| Data exclusions | No data were excluded from the analysis. |
| Replication | Reproducibility and reliability of the findings has been ensured in several ways. In most cases, biochemical experiments in vitro and functional assays in cells were performed on separate and fully independent occasions and verified to give the same result as the example shown in the figure. All experiments are performed at least twice. |
| Randomization | Sample randomization is not relevant to our functional experiments because we routinely verify the identify of cells were used in our experiments. Hence, samples were not randomized for the functional assays because there is nothing to randomize. The different conditions being compared within any given experiment derive from a single common stock of reagent or a single culture of cells, so random assignment or covariates are not relevant to this type of study. |
| Blinding | Blinding is not performed in functional assays because the experiments have internal or external controls to indicate the identify of the cells we were assaying. |

# Reporting for specific materials, systems and methods

We require information from authors about some types of materials, experimental systems and methods used in many studies. Here, indicate whether each material, system or method listed is relevant to your study. If you are not sure if a list item applies to your research, read the appropriate section before selecting a response.

## Materials & experimental systems

| n/a | Involved in the study |
|---|---|
| ☐ | ☒ Antibodies |
| ☐ | ☒ Eukaryotic cell lines |
| ☒ | ☐ Palaeontology and archaeology |
| ☒ | ☐ Animals and other organisms |
| ☒ | ☐ Clinical data |
| ☒ | ☐ Dual use research of concern |
| ☒ | ☐ Plants |

## Methods

| n/a | Involved in the study |
|---|---|
| ☒ | ☐ ChIP-seq |
| ☐ | ☒ Flow cytometry |
| ☒ | ☐ MRI-based neuroimaging |

## Antibodies

| | |
|---|---|
| Antibodies used | CCDC47 (Bethyl Laboratories A305-100A, 1:5000); EMC3 (Invitrogen 711771, 1:5000); EMC6 (Abcam ab84902, 1:1000); Calnexin (Enzo ADI-SPA-865, 1:5000); Sec61α (ref70, 1:5000); TMCO1 (Invitrogen PA5-43350, 1:500); Sec61β (ref71, 1:5000); CAML (Cell Signaling Technology 13913S, 1:1000); FLAG M2-HRP (Sigma A8592, 1:5000); EMC4 (Abcam, ab123719, 1:2000); β-Actin-HRP (Sigma, A3854, 1:10000); phycoerythrin (PE) labelled FLAG antibody (BioLegend 637310, 1:100); GABRA1 antibody (Invitrogen, PA5-79291, 1.25µg per IP). |
| Validation | Each antibody was validated for specificity against the antigen by the manufacturer as follows: <br> phycoerythrin (PE) labelled anti-FLAG Tag Antibody: Validated for surface labeling by manufacturer (BioLegend). Species not applicable here because FLAG is an epitope tag that has no species. <br> Monoclonal ANTI-FLAG® M2-Peroxidase (HRP) antibody produced in mouse: validated by manufacturer (Sigma) for IB. Species not applicable here because FLAG is an epitope tag that has no species. <br> Anti-EMC4 antibody: validated for IB against human protein by manufacturer (Abcam) against human protein. <br> Anti-β-Actin–Peroxidase antibody, Mouse monoclonal: validated by manufacturer (Sigma) for IB against human protein. <br> GABRA1 antibody: validated by manufacturer (Invitrogen) for IB against human and mouse proteins. <br> CCDC47 antibody: validated by manufacturer (Bethyl Laboratories) for IB against human protein <br> EMC3 antibody: validated by manufacturer (Invitrogen) for IB against human protein. <br> EMC6 antibody: validated by manufacturer (Abcam) for IB against human protein. <br> Calnexin antibody: validated by manufacturer (Enzo) for IB against human protein. <br> TMCO1 antibody: validated by manufacturer (Invitrogen) for IB against human protein. <br> CAML antibody: validated by manufacturer (CST) for IB against human protein. <br> Sec61α: validated by previous work (Song et al., 2000) for IB against human protein. <br> Sec61β: validated by previous work (Fons et al., 2003) for IB against human protein. |

## Eukaryotic cell lines

Policy information about cell lines and Sex and Gender in Research

| | |
|---|---|
| Cell line source(s) | HEK293 FRT/TO TRex cells were originally purchased from Invitrogen (R78007). <br> ΔEMC6 Flp-In™ T-REx™ 293 cells were described before (Guna et al., 2017) and were obtained from the Medical Research Council Laboratory of Molecular Biology (MRC-LMB). <br> ΔTMCO1 293 cell line was a gift from Robert Keenan (University of Chicago). <br> ΔEMC6ΔTMCO1 293 cell line was generated in this study. <br> Flp-In™ T-REx™ 293 cells stably expressing wild type or EMC3-FLAG mutants were described before (Wu and Hegde 2023) and were obtained from MRC-LMB. <br> Tetracycline-inducible HEK293 cell line expressing human GABAA receptor (alpha subunit is FLAG tagged at the N-terminus; beta subunit is untagged; gamma subunit is 1D4 tagged at the C-terminus) has been described before (Dostalova et al., 2014) and were obtained from MRC-LMB. <br> Flp-In™ T-REx™ 293 cell line stably expressing GFP-P2A-RFP-ASGR1 or GFP-P2A-RFP-SQS were described before (Chitwood et al., 2018) and were obtained from MRC-LMB. |
| Authentication | Cell lines were not authenticated beyond ensuring the presence of known antibiotic resistance markers within their genomes (by growth in the relevant antibiotics) and by their unique FRT site downstream of a doxycycline-inducible promoter as determined by the ability to integrate fluorescent reporters at this site. |
| Mycoplasma contamination | Cell lines were negative for mycoplasma. They are tested monthly. |
| Commonly misidentified lines (See ICLAC register) | None used. |

# Flow Cytometry

## Plots

Confirm that:

☒ The axis labels state the marker and fluorochrome used (e.g. CD4-FITC).

☒ The axis scales are clearly visible. Include numbers along axes only for bottom left plot of group (a 'group' is an analysis of identical markers).

☒ All plots are contour plots with outliers or pseudocolor plots.

☒ A numerical value for number of cells or percentage (with statistics) is provided.

## Methodology

| | |
|---|---|
| Sample preparation | Samples consisted of HEK293-derived cell lines that stably expressed a fluorescent protein reporter or GABAA receptor for monitoring surface expression. Where indicated in the Methods, they were first treated with siRNAs. The cells were collected in ice-cold PBS, washed and resuspended in PBS supplemented with 2% FCS and 1 µg/ml DAPI (Thermo Fisher Scientific). Where indicated, surface antibody/GABAA receptor labeling is performed after pelleting cells. Cells were passed through 70-µm filter immediately prior to analysis using Beckton Dickinson LSRII or LSRFortessa instrument. A total of at least 30,000 fluorescent and live (negative for DAPI stain) cells were collected. |
| Instrument | Beckton Dickinson LSRII or LSRFortessa. |
| Software | FlowJo (version 10.8.0). |
| Cell population abundance | A total of at least 30,000 live cells (negative for DAPI stain) that also were positive for the fluorescent protein reporter (either PE, GFP or RFP) were analyzed. |
| Gating strategy | Gating was used only to include cells, rather than debris. Further gating was not used. |

☒ Tick this box to confirm that a figure exemplifying the gating strategy is provided in the Supplementary Information.

