## [Peer Review File · Nature Structural & Molecular Biology]

Peer Review Information

Manuscript Title: EMC rectifies the topology of multipass membrane proteins

Corresponding author name(s): Ramanujan Hegde

Reviewer Comments & Decisions:

Decision Letter, initial version:

Message: 30th May 2023

Dear Dr. Hegde,

Thank you again for submitting your manuscript "EMC rectifies the topology of multipass membrane proteins". We now have comments (below) from the 2 reviewers who evaluated your paper. In light of those reports, we remain interested in your study and would like to see your response to the comments of the referees, in the form of a revised manuscript.

You will see that while the reviewers are overall positive, they raise several concerns which will need to be addressed in a revision. Specifically, we would ask that descriptions of methodologies are provided in more detail, particularly of the constructs used in the study, in line with reviewer's #1 comments. Furthermore, we would expect the experimental data to be quantified and requested controls added, as suggested by reviewer #2, and aspects of tail translocation analysis revisited to include normalisation, as suggested by reviewer #1. Reviewer #2 proposes that other factors could play a role in C-terminal TMD insertion – we agree that further investigation of involvement of these would strengthen the manuscript, and should be explored if feasible. The caveats of the experimental system used should be discussed at the very least.

Please be sure to address/respond to all concerns of the referees in full in a point-by-point response and highlight all changes in the revised manuscript text file. If you have comments that are intended for editors only, please include those in a separate cover letter.

We expect to see your revised manuscript within 12 weeks. If you cannot send it within this time, please contact us to discuss an extension; we would still consider your revision, provided that no similar work has been accepted for publication at NSMB or published elsewhere.

Reporting Summary:

Please note that all key data shown in the main figures as cropped gels or blots must be presented in uncropped form, with molecular weight markers. These data can be aggregated into a single supplementary figure item. While these data can be displayed in a relatively informal style, they must refer back to the relevant figures. These data should be submitted with the revision, as source data.

SOURCE DATA: we request authors to provide, in tabular form, the data underlying the graphical representations used in figures. This is to further increase transparency in data reporting, as detailed in this editorial (<http://www.nature.com/nsmb/journal/v22/n10/full/nsmb.3110.html>). Spreadsheets can be submitted in excel format. Only one (1) file per figure is permitted; thus, for multi-paneled figures, the source data for each panel should be clearly labeled in the Excel file; alternately the data can be provided as multiple, clearly labeled sheets in an Excel file. When submitting files, the title field should indicate which figure the source data pertains to. We encourage our authors to provide source data at the revision stage, so that they are part of the peer-review process.

Data availability: this journal strongly supports public availability of data. All data used in accepted papers should be available via a public data repository, or alternatively, as Supplementary Information. If data can only be shared on request, please explain why in your Data Availability Statement, and also in the correspondence with your editor. Please note that for some data types, deposition in a public repository is mandatory - more information on our data deposition policies and available repositories can be found below: <https://www.nature.com/nature-research/editorial-policies/reporting-standards#availability-of-data>

[redacted]

Sincerely,
Kat

Katarzyna Ciazynska
(she/her)

Associate Editor
Nature Structural & Molecular Biology
<https://orcid.org/0000-0002-9899-2428>

Referee expertise:

Referee #1: protein biogenesis, biochemistry

Referee #2: ER protein translocation

Reviewers' Comments:

Reviewer #1:

Remarks to the Author:

This manuscript uses a range of biochemical methods to elucidate the role of the endoplasmic reticulum membrane protein complex (EMC) during insertion of multipass membrane proteins into the endoplasmic reticulum. The authors identify the EMC as an important complex for inserting the final transmembrane domain (TMD) of multipass membrane proteins when situated within approximately 50 amino acids of the C-terminus. This finding carries implications for an estimated 250 multipass membrane proteins within the human genome that meet these criteria. The presented data is of good quality and the manuscript is well structured and easy to read. Nevertheless, some of the methodologies and constructs should have been described in more detail. Additionally, the authors should address the following points.

Major comments:

1. In Figure 2B, the knockdown of EMC using siRNA appears to be nearly complete. However, there seems to be a discrepancy when compared with the surface levels of GABAAR in these cells, which are reduced to approximately 50% as shown in Figure 2A and described in the text. Is it possible to explain this inconsistency considering the comments on page 4 indicating that the C-terminal TMD insertion process is highly inefficient when unassisted? Is it possible that the process involves alternative or backup insertion mechanisms?
2. A detailed description of the construct used for the overexpression of the GABAA receptor should be provided. Specifically, the precise location of the FLAG-tag (used for surface labeling) and its relation to the signal sequence warrant further elaboration. Citing another study (which in turn references yet another study for specifics) is cumbersome and frustrating to the reader.
3. In the immunoprecipitation (IP) experiments, faint bands indicative of crosslinking products with EMC3 can be observed, however the bands indicated by the green arrows in Figure 3B are difficult to discern. Is it possible to provide a more convincing gel? Would it help to provide a densitometry reading across the entire lane?
4. Page 5, line 14: "Charged residues in the C-terminal tail of 23L-SQS were seen to modestly but clearly reduce tail translocation (Fig. 4b)." However, it is difficult to discern the described reduction in tail translocation for WT SPCs as depicted in Figure 4B. The authors should provide a densitometry reading represented as a bar graph, normalized to 23L-SQS variant "0", which could offer a better way to appreciate the signal reduction.

5. The list of ~250 potential EMC substrates obtained by manual curation of all ~1800 multipass ER membrane proteins in human genome should be provided as supplementary data. This should provide a useful resource for future research in the field.

Minor comments:

1. EMC full name (endoplasmic reticulum membrane protein complex) is not mentioned either in the abstract or in the main text, please expand the abbreviation first time it is mentioned.
2. It would be beneficial to include a depiction of the penultimate transmembrane domain (TMD) in Figure 1C. This addition would ensure consistency across the panels in Figure 1 (since panels A and B already display both the penultimate and the last TMD) and improve the figure in the context of the biogenesis scenarios described in the text.
3. The authors should refer to the figure 1C in the penultimate sentence of the introduction when explaining the TMD4 insertion of GABAA receptor.
4. Figure 4, please indicate in the figure legend that ΔE equals ΔEMC .
5. There are references to Fig. 5a and Fig. 5b in the main text, but figure 5 has only a single, unlabeled panel. Most likely the figure labels are missing in figure 5.

Reviewer #2:

Remarks to the Author:

Unlike the translocation of soluble secretory proteins, the process of integrating multipass transmembrane proteins is substantially more complex, as the factors mediating the process vary depending on the client protein's distinct topological requirements. In this manuscript, Wu and Hegde investigated the role of the EMC complex in the integration of TMDs close to the C-terminus. They proposed a model in which these TMDs insert into the membrane post-translationally as they are released from the ribosome before engaging with the Sec61 channel, and the insertion process is mediated by the EMC complex, resembling the mechanism observed in TA protein insertion. These steps in insertion of multipass membrane proteins have received limited prior investigation, and thus in my view, the findings are exciting. Overall, the presented data is of high quality, and the model is sound. While I would support publication of this work in NSMB, I have some concerns regarding the quantitative aspects of the data and its interpretation.

1. The entire study is focused exclusively on the EMC complex. No other alternative mechanisms or players were considered. In the TA protein targeting pathways, Get1/2 also serves as a similar insertase. Additionally, as the authors indicated in the introduction, the GEL complex is also related to the EMC, which raises the possibility that GEL is involved in this process. While the current data suggest involvement of EMC in the process, it remains unclear whether EMC acts the sole or central player. Looking at the data presented in Fig. 1, it appears that EMD KO only causes at most a 50% reduction, indicating that the remaining 50% of the model client proteins insert correctly without EMC, potentially through spontaneous insertion or with assistance of another factor, such as GET, GEL, and Sec61. Without negative data showing minimal involvements of these other factors (using KD or KO), the authors' conclusions could be overinterpretation (for example, the abstract states "requiring" a separate EMC-mediated post-translational insertion step").

2. Given the significance of EMC in ER protein biosynthesis, it is possible that KO and depletion influence the levels of other proteins involved in protein biogenesis, such as GET, GEL, and Sec61, and this has indirectly affected the insertion efficiency of the model

TMDs. Thus, it is necessary to confirm that the levels of these proteins remained unchanged following EMC KO and KD.

3. While certain data have been quantified, other data were not (for example, Figs. 1c, 1b, 2c, and 4). It is also unclear how many times these experiments are replicated (including Figs. 5 and 6c). Considering the relatively modest effects observed with EMC KD and KO, it is necessary to include quantification and statistical tests. Also, the figure legends and Methods section need detailed descriptions on how band intensities are quantified and normalized.

4. I could not understand the rationale behind using the Mcyt1-S and R31A mutants in Figs. 1e and 2a, considering that their effects are even more moderate with only a 25-30% decrease compared to WT. Is this because the mutants are only partially impaired or because there is some residual WT protein in the cells? I don't see how these data contribute to supporting the authors' conclusions.

5. Fig 2 b and c: These need an additional negative control besides EMC KO (the absence of the crosslink band here is rather trivial) showing the crosslinking is specific to the EMD's activity. This is probably a good place to use Mcyt1-S and R31A. Related to point 1, the authors could also test crosslinking to other potential players, such as GET, GEL, and Sec61, to strengthen their conclusions (I believe this is feasible with Bpa crosslinking).

6. Fig 4b: First, in order to compare the insertion efficiencies between a TA protein and a multipass membrane protein, it is important for the authors to include the same set of experiments done with the TA version. Although similar experiments might have been performed previously, there could be some differences in the experimental setups. Thus, an equivalent set for the TA version is necessary. Secondly, a recent JCB paper by Pleiner et al. showed that +2 and +3 charges only moderately affect the SQS insertion efficiency (particularly used in in-vitro translation). The authors should consider testing +4 and +5 before making the conclusion.

7. Fig 6c: This data does not convince this reviewer. In the case of SOAT1, the +glyc band is very diffuse, making it unclear whether the band intensity can be reliably quantified. In addition, YipF1 showed relatively only marginal effects. To support the authors' model, this section would need additional quantitative methods, such as flow cytometry, as well as testing more client proteins from the list (also provide the list in supplementary information).

8. In my view, the current version of the Methods section is not detailed enough. The authors should include more detailed information about protein sequences of constructs, sources of reagents, amounts/volumes of reactions, and so on. Also, there is no specific information on how plots (list of candidates) in Fig 6 were generated.

Author Rebuttal to Initial comments

We have revised our submission in response to all reviewer comments. We have shortened and streamlined multiple sections, have moved a number of figures to supplement, and removed others entirely. Most importantly, we have added substantial new data in three areas. First, we measure NTP levels in Pol II mutants, finding one slow mutant perturbs NTP levels, potentially explaining buffering of +1G effects in this mutant relative to +1A effects. In the fast mutant, we do not observe NTP level changes, supporting direct effect of this mutant on selective effects for +1A relative to +1G. Second, we demonstrate huge effects of reduced GTP levels on +1G TSS efficiency across all sites as measured by Pol II MASTER. Third, we find global changes to TSS efficiencies in the genome in response to altered NTP concentrations in vivo. Specifically, these results indicate that GTP limitation can affect initiation efficiency due to G position within the first five nucleotides of transcripts. We also find evidence that increased CTP and UTP also increase initiation efficiency for TSSs that have C/U at the second position. These results indicate that initiation by promoter scanning is widely sensitive to NTP substrate levels. Our point-by-point responses are below in blue italics.

Reviewer #1

In this manuscript, Zhu and colleagues developed Pol II MASTER system based on bacterial MASTER (Vvedenskaya et al 2015) and applied it to quantify transcription initiation efficiency for ~80,000 promoter variants in both wildtype and Pol II mutants. Pol II MASTER uses transcription initiation signals in Flux Detector Region as a reference to infer relative TSS efficiency, which is similar to the use of tandemly duplicated start sites in (Kuehner & Brow, 2006). The main findings of this study confirm the critical roles of +1, -1 and -8 sites in TSS selection and it also demonstrates interactions between some adjacent sites in TSS regions.

I think this paper presents an important work with two major contributions to the field. First, the quantitative data is valuable for better understanding the roles of TSS-proximal sequences on its transcription efficiency, which is instrumental for developing a predictive model of TSS efficiency (this study). It will be also useful for developing TSS prediction model based on genomic sequence and studies of the genetic basis of different core promoter shape (distribution of TSS signals in a core promoter). Second, although the development of Pol II MASTER in this study is not entirely original, it can serve as a powerful tool for functional characterization of other sequence elements related to transcription initiation. I think this paper can be further strengthened by some tests on the robustness of the MASTER system and data analysis.

We thank the reviewer for their positive comments and careful review. We provide data here that our analyses are especially robust. In a figure provided here for the reviewer, we provide DNA-template normalized analysis. This analysis indicates that there is no strong interaction between upstream positions and the downstream “flux detector” TSS (see figure below, with caveats noted). More importantly, we now present new data indicating that initiation is sensitive to altered NTP levels in vivo, using both our MASTER libraries and examination of genomic TSSs upon treatment of cells with mycophenolic acid (which alters NTP levels). We also find that altered NTP levels are likely part of the physiology of “slow” Pol II mutants. This effect could be due to their known constitutive expression of IMD2 - the rate limiting step in GTP synthesis as well as URA2 and potentially other genes.

These are major new results in addition to the additional analyses we have done in response to review.

Major suggestions and comments :

1. Most of the conclusions in this study rely on the calculation of TSS efficiency, which is the relative portion of TSS signal to its downstream region (or Flux Detector region for +1 TSS). I understand it is a way to normalize TSS signals to make them comparable between variants. This is based on assumptions that the total Pol II flux is the same among variants and FD regions has no impact on transcription from upstream sites. Since the authors did both DNA-seq and TSS-seq, it is possible to infer the strength of transcription initiation from a site as the number of TSS-seq reads (or normalized counts) per # of DNA (or sequence depth) for each variant (similar to #match reads/#DNA in Table S1 in Vvedenskaya et al 2015) or their log₂ ratios to the variant with wild-type TSS sequence. I don't think it will change its major qualitative conclusions, but consistency of quantitative results would support the robustness of analytical methods used in this study, which is important for developing a more accurate predictive model. For example, the authors observed different hierarchies of efficiency of YR elements between libraries and previous genomic studies. Could such discrepancy be also observed by using a different calculation approach?

Previous genomic studies from others and our own have examined preference for start sites based on usage either relative to total amount observed or normalized to expression bins. Because scanning is a “first come-first served” situation, priority effects require the position of TSS to be accounted for to enable comparison of different sequences in different contexts. If we look at measured TSS efficiency vs template normalized usage (relative expression), we find no strong evidence of TSS region interfering or altering total flux (see figure below). Interaction between +1 TSS and FD potentially would be observed as a change in overall expression of a promoter (determined by normalization of RNA counts to DNA counts) correlating with +1 TSS behavior. In contrast, we observe that overall promoter expression seems consistent across a range of +1 TSS efficiencies. We note that this is still possible for a subset of TSSs, but our experiments are not designed to be able to detect this. This is due to confounding effects of requiring a barcode within the transcribed region.

Relative expression of promoter v.s. efficiency of major TSS

2. Given that it is possible to infer TSS efficiency independent of FD region, it raises a question of whether adding FD is necessary or what is the potential impact for having a strong TSS region downstream of randomized region. In Kuehner & Brow 2006's study, RNA levels from a TSS were quantitated based on gel image, so a strong downstream TSS region was used as an internal control. If its upstream TSS region is also highly efficient (i.e., AYR promoters), the presence of a highly efficient TSS region immediately downstream of it may not noticeably impact upstream transcription initiation activities according to the unidirectional scanning model. However, its impacts on less efficient upstream promoters (i.e. non-AYR) need to be more carefully evaluated (interference or competing effect?). For example, Fig. 1D shows higher fractions of TSS usages in the weak promoters (ARY). The data in Table S1 in Vvedenskaya et al. 2015 show three orders of magnitude difference in #match reads/#DNA among variants. It would be interesting to learn the extent to which transcription initiation activities/efficiency change without a downstream efficient TSS region, or with an unfavorable sequence. Construction of new libraries is probably too much work, but some comparative analyses with results in Vvedenskaya et al. 2015 Discussion would be helpful.

As noted above, analysis of relative expression (taking into account template levels based on DNA sequencing) vs major TSS efficiency suggests that FD doesn't strongly affect upstream TSSs, consistent with a priority model for scanning and independence of TSSs at least in our designed promoter context. We note the caveat that we can't rule out effects of a small subset of sequences as an additional variable in the experiment is that the transcribed regions must be barcoded and barcodes could conceivably alter mRNA properties.

3. The study of -9/-8 interaction was based on promoter variants with a TSS at +4, which have the same dinucleotide (CA) at TSS in all sequences. I am curious if their mutual A suppression is specific to TSS with CA? As -8/-7 positions are within the randomized region, did the authors use the entire library, or were the same subset sequences used for -9/-8 interaction?

Because of the design of libraries, this can't be addressed based on libraries, because the +4 TSS of (AYR+BYR) is the only dataset that has N-9N-8. The fact that we can observe this effect on genomic TSSs suggests that the effect goes beyond CA (see figure below). The figure indicates when the -9 position is an A, the preference for A at +1 is reduced; similarly, when -8 is an A, the preference for -9A is reduced and this is not dependent on C-1A+1.

Other comments:

1. I'd recommend adding nucleotide sequences (or at least sequences of flux detector region) under x-axis in Fig 1D. For example, the shift of TSS from +1 to +2 in the "ARY" group is because +2 is A, and replacing R with Y at +1 site (Y-1R+1A+2 -> R-1Y+1A+2) creates a new YR site. With sequence includes, it provides intuitive visualizations for the critical role of YR elements in transcription initiation.

Figure 1D altered as requested.

2. It is not clear to me whether the authors examined all combinations of 9 bp randomized TSS regions ($4^9=262144$) or just a portion of them (AYR+BYR+ARY ~ 80,000 promoter variants). In addition, the authors used "1,000,000 individual TSS sequences" in the abstract (by including TSS sites from +4 site). I understand that the definition of "TSS sequence" is different from "promoter variant" here, but it was confusing. Some clarification would be helpful.

Thanks for this comment. We have attempted to clarify that promoter variants (~80K) can have TSSs at multiple positions with them, allowing interrogation of ~1M distinct sequences.

3. I think it would be helpful if the authors provide a supplementary table that contains numbers of DNA-seq reads, TSS-seq reads mapped to each position for each variant.

Good suggestion. We had these types of details in our GEO submission but have added some as supplemental here.

4. The function of -8A and A-rich region as an anchor point for PIC scanning was previously proposed in (Lu and Lin 2021).

Thanks for catching this. We absolutely want to make sure Lu and Lin are referenced.

Reviewer #2

In this work, the authors investigate sequence preferences for yeast transcription start sites using a randomized promoter library and sophisticated computational analysis. Their analysis confirms the previously reported importance of DNA sequence identities at position -8 (that likely interacts with TFIIIB in the PIC) and the preferred sequences surrounding the TSS (-1 and +1 positions) that seem universal for all RNA Pols. They also identified a new base preference at position -9, and weak preferences just upstream and downstream of the TSS. Finally, they developed a TSS predictor using a regression model that performs best on predicting TSS usage at highly expressed genes. While this is generally all very nice, my main problems with this manuscript are (i) the manuscript and figures are very difficult to read making it very frustrating to try to read carefully through the manuscript and (ii) it was disappointing that the data and analysis didn't lead to significant mechanistic insights into transcription initiation. In my view, the value of the predictor and associated data lies in its potential to say something important about the transcription initiation mechanism. It's relatively easy to map TSSs genome-wide so, in that sense, a predictor is not necessary. However, if one could work backwards from these sequence preferences and the predictor to say something important about how Pols initiate it would be very valuable. Unfortunately, new mechanistic insights were not apparent from the manuscript and new data. I think that the manuscript is much more appropriate for a specialized journal, especially in light of point (ii).

We appreciate the critical reading of our manuscript. Our manuscript provides several mechanistic insights. First, we detail quantitatively how sequence contributes to Pol II initiation by scanning and show that positions outside of -8,-1,+1 can tune initiation over wide ranges of efficiency. Second, we present evidence that the Pol II active site can selectively alter sequence effects on initiation by scanning through the +1 and downstream positions. We present evidence that suggests that Pol II catalytic mutants do not affect initiation through the -8 position, in contrast to conclusions that would be made from observed usage changes at genome promoters in vivo. We discover a -9 interaction with -8, setting stage for these positions to be tested in conjunction with TFIIIB mutants to specifically test that proposed mechanism in future studies. Most importantly, as noted above in our summary, we have added new data indicating that initiation is globally sensitive to NTP levels and TSS efficiency is sensitive to multiple NTPs at multiple positions within early initiation. We also provide evidence that one Pol II mutant, Pol II E1103G, appears to selectively alter efficiency at the +1 position, while another, Pol II F1086S, may do so indirectly through altered NTP levels. These are major new conclusions indicating a cryptic regulatory layer for the amounts of individual transcripts with the potential to act on all transcripts. Prior work in yeast has strongly predicted that one promoter, IMD2, might have evolved to directly sense reduction in GTP levels. Our work here indicates that this is not a unique or specialized context and that all transcripts containing Gs within the initial transcribed region have potential to be affected by altered GTP levels. We also find evidence that increased CTP and UTP (a consequence of our manipulation of GTP levels with mycophenolic acid) can increase efficiency of TSSs with C or U at +2.

I suggest that this manuscript should be revised to make it more accessible and to consider the following suggestions:

1. I had a major problem with the authors definition of transcription efficiency which the authors use repeatedly as a measure of TSS activity. They don't explain what exactly they mean by this term when first introduced, and it's not easily apparent how it was defined in reference (22). Finally, the methods section was of little help on this point. How this is calculated need to be much better explained and justified in the main text.

We make sure that we discuss this concept, and why it is essential for the comparison of TSSs, in the introduction and have now added additional clarification in results. We have also clarified this in methods.

2. There are way too many figures and this very much distracts from the important points of the paper in my opinion. It's not necessary to show every single analysis that was done - show only a subset of the most important experiments and analyses. This will also make the work accessible to more than a handful of experts.

We have removed a number of figures to supplemental material and attempted to move some of the discussion and presentation there as well. We have also removed some supplemental figures.

3. The section on Pol II mutants and TSS usage is very hard to understand. Part of the problem is Fig S4A: Wild type in grey is invisible so one can't compare TSS usage of the mutant with WT. Also the format of the heatmap in Fig S4J (repeated in other figures) is impossible for me to decipher what is being shown.

We apologize for the lack of clarity here. We have adjusted figure size here and have clarified in figure legends what heat maps represent. These heatmaps indicate where the base at one position shows an interaction with a base at another position. Each position can be one of 4 bases. Therefore, a 4x4 matrix indicates the potential interactions between one position and another. If the effect on TSS efficiency of a base at one position is coupled or interacts with a base at another position, the intersection of the x and y axes will indicate if that particular sequence combination deviates from the efficiencies of all sequences. For example, if there is an A at -9, there is a relative decrease in preference for an A at -8. This is indicated by a negative value (blue) in the heatmap at the intersection of -8A and -9A.

4. I didn't understand the concept of "Median TSS" (line 410). Is this an actual TSS or a window containing a certain fraction of TSSs? What is the justification for this?

We apologize for the lack of clarity here. This analysis simply utilizes one TSS per promoter window to assess the sequence model derived from our promoter library. The "median TSS" is the TSS that contains the 50%ile of reads distributed across a promoter window. In the cumulative distribution function of TSSs in a genomic region, one position will mark the

spot where half the reads are upstream of that position and half are downstream. We describe this in a few places of the revised manuscript. First, in the introduction at line 96 and second, in Methods “TSS-seq analysis for genomic TSSs” (starting line 847).

Reviewer #3

In “Quantitative analysis of transcription start site selection in *Saccharomyces cerevisiae* determines contributions of DNA sequence and RNA Polymerase II activity”, the authors describe a new parallel reporter assay technology, Pol II MASTER, and use it to measure sequence parameters of transcriptional start site selection across a library of ~250,000 barcoded sequence variants within a ~20 bp window upstream and downstream of the TSS. By integrating across their library, they identify positions -9, -8, and -4 to +1 as impactful on TSS selection efficiency, recapitulating known TSS selection impacts of the -8, -1, and +1 positions, and identifying a particular coupling between the -9 and -8 nucleotides. The authors also investigate the interplay of sequence variants and Pol II mutants, finding the effects of these mutants to be largely sequence independent, with the exception of specific +1 nucleotide preferences. They also develop a regression model that accurately predicts TSS efficiency, incorporating nucleotide information from nine positions, and suggest that simple nucleotide composition at specific locations, rather than complex multi-position interactions, can explain the bulk of the variation in TSS efficiency.

Throughout the work, the authors make three general claims: (1) that the Pol II MASTER methodology is capable of identifying sequence features that control TSS selection; (2) that Pol II mutants have specific preferences for the initiating nucleotide, but do not otherwise demonstrate sequence specificity; and (3) that a functional model of TSS activity incorporating the nucleotide composition of nine positions surrounding the TSS is capable of explaining the bulk of the observed behavior in their system.

In general, the described experiments are rigorous and broadly support the proposed claims. Included below are several comments on the work:

Based on the data presented in Figure 1D, and Figure S1D, the replicate-to-replicate reproducibility of Pol II MASTER measurements yields an r of ~0.9. However, in Figure 6C, the R^2 correlation reported between the predicted and measured efficiency of the regression model is reported as 0.91. The fact that the performance of the regression modeling appears to exceed the inter-replicate measurement reproducibility suggests that the model may be over-fit.

We repeated our modeling using the individual replicates of WT and Pol II mutants. Parameters for replicates of each genotype cluster more closely than the parameters for the different Pol II mutants. This is demonstrated by PCA of the individually modeled replicates (see Figure S5C). This suggests that we are detecting real biological differences and models are not overfit. We also note that by fitting the combined data from all replicates there is expectation that values determined by all replicates would be more accurate than those determined from individual replicates, and thus an accurate model could have R^2 in the range of the r for replicates. Another key idea is that the model derives from a huge number of sequence variants fitted by a vastly smaller number of

parameters. We discussed this with a modeling expert Justin Kinney from CSHL and there was not a concern about overfitting.

As a whole, the manuscript is overly dense and portions of the text are repetitive or contain extraneous detail. Condensing the text to material specifically relevant to the current work would go far to improve readability. By way of example, lines 52-56 of the introduction, or lines 135-138 of the results, while informative, do not contribute to the interpretation of this work; and these are only two instances of multiple throughout the text.

We apologize for this. Our goal was to be as precise as possible. We have shortened the manuscript by multiple pages and streamlined a number of sections. This is offset by inclusion of new data. A number of figures have been removed to supplemental or altogether.

Minor Comments:

The discussion section would likely benefit from copy-editing; for instance, lines 515, 532, 587

***Discussion has been revised for many paragraphs, but we note specifically:
Prior line 515, current 502-504, we fixed a missing space and revised:***

We propose that two major sequence position groups contribute to TSS selection: bases around the TSS and bases around position -8.

Prior line 532 has been completely revised. Prior line at 587 has been revised as:

Whether or how DNA sequence surrounding TSSs is involved in other promoter properties is another question. A bases at positions -7 to -5 were observed to be neutral in our promoter libraries (Figure 2E), in contrast to A-enrichment observed for highly expressed and focused genomic TSSs (Figure 5A, B)

Figure S4, which spans three pages of the compiled manuscript, should be broken into multiple component figures

We have removed some supplemental figures and reorganized figures. One or two still span more than a page but none span three pages.

Decision Letter, first revision:

Message: Our ref: NSMB-A47639A

11th Aug 2023

Dear Dr. Hegde,

Thank you for submitting your revised manuscript "EMC rectifies the topology of multipass membrane proteins" (NSMB-A47639A). It has now been seen by the original referees and their comments are below. The reviewers find that the paper has improved in revision, and therefore we'll be happy in principle to publish it in Nature Structural & Molecular Biology, pending minor textual revisions to satisfy the referees' final requests and to comply with our editorial and formatting guidelines.

To facilitate our work at this stage, it is important that we have a copy of the main text as a word file. If you could please send along a word version of this file as soon as possible, we would greatly appreciate it; please make sure to copy the NSMB account (cc'ed above).

Sincerely,

Katarzyna Ciazynska
(she/her)
Associate Editor
Nature Structural & Molecular Biology
<https://orcid.org/0000-0002-9899-2428>

Reviewer #1 (Remarks to the Author):

The authors have adequately addressed the reviewers comments.

Reviewer #2 (Remarks to the Author):

The revised manuscript by Wu et al somewhat improved the original version, particularly with additional tests for putative EMD-dependent client proteins and some clarifications. While overall the presented data seems to consistently indicate decent contributions of EMD to membrane insertion of the last TM of these proteins, I am not fully convinced with

some of the authors' data and interpretations. Although I still think this study provides important insights, I believe some additional data and textual revision would be necessary before publication.

1. In the revised manuscript, the authors tested six additional potential client proteins for enhancement of their membrane insertion by EMD using semi-permeable cells. However, these constructs are chimeric constructs with just their terminal TM fused to 23L, not the native proteins. I believe that a simple way to provide evidence for EMD dependency would be just comparing steady state expression levels of full-length proteins (whether endogenous or ectopically overexpressed) in the cells with and without EMC as is done with GABAR in Fig. 2b.

2. The overall contribution of EMC in this process seems to be at best ~50% (which I certainly think is still significant). I agree with the authors that the remaining activity might be through spontaneous insertion or unknown factors. In some cases, such as CNIH2, S38A1, and YIPF1, EMC's contribution seems quite marginal too. On the other hand, many statements throughout the text were written in assertive manners. For example, in the abstract, the authors state "we find that TMDs close to the C-terminus of a multipass protein are inserted post-translationally by the ER membrane protein complex (EMC)." This is quite a general statement, and it is not proper for a 50% contribution. Like the authors acknowledged in their rebuttal, "facilitates" would be more reasonable. Other examples of overgeneralized statements are:

- a. "... requiring a separate EMC-mediated post-translational insertion step to rectify their topology" (abstract).
- b. "its lack of surface expression... a failure in biogenesis" (page 3).
- c. "C-terminal translocation of 23L-GABRA1 was EMC dependent" (page 4).
- d. "Nonetheless, such alternative mechanisms seem to be minor contributors relative to EMC" (page4).
- e. "We conclude that the last TMD of GABRA1 is post-translationally inserted into the ER membrane through EMC" (page 5).

3. In page 7, "Thus, EMC's substrate preference against positive charge translocation can be overcome by simply providing more time, meaning that it can accommodate a broader range of substrates as a terminal TMD insertase." Although I agree that this is a plausible explanation, I think this interpretation is somewhat speculative. The statement needs to be toned down.

4. In page 9, "... can now be ascribed to a failure of terminal TMD insertion." This statement is also too strong and needs to be toned down. For GABRA1 and SOAT1, the decrease is only ~50%, and it is entirely unclear what percentage of this is due to EMC's insertase activity versus putative chaperonic function. Both GABRA1 and SOAT1 are multimeric membrane proteins, and EMC's chaperonic function could be an important part of their proper folding. The current mutant data (Fig 1e) is not quite definitive in distinguishing these two activities (considering partial reduction). Alternatively, could the authors provide data that the EMC mutants used in Fig. 1e also show similar defects in protein expression levels (as in Fig. 1a,b)?

5. In page 5, "... not some other part of EMC such as its putative chaperone surface." This is related to the above point. I am not sure how the authors can exclude this possibility given that the effects of Mcys-1-S and R31A are partial (weaker than EMD KO/KD).

6. In page 20, the authors should be more specific about how TMD helices (start and end positions) were annotated. Are these just roughly picked by the authors or using some systematic rules? Also, it would be more helpful to readers if the authors include the sequences of the last TMD and C-terminal tail in the Excel table.

Author Rebuttal, first revision:

Revision 2 response:

We thank the reviewers for their comments and critiques. We have added our comments below (inline and in italics).

Reviewer #1:

Remarks to the Author:

Based on the authors' responses to the comments of reviewers and revisions in the manuscript, I believe several major concerns raised in the reviews have not yet been satisfactorily addressed.

1. Responding to the first issue in my original review, the authors provided a figure of relative expression vs TSS efficiency to show that promoter expression is consistent with TSS efficiency. However, the figure does show noticeable variations in promoter expression levels (many $> 1 \log_2$ fold changes), and the distributions of relative expression are uneven among different TSS efficiencies.

We apologize for the confusion and have updated our figure for review and have added a version of these figures to the manuscript as Figure S1H. Because we neglected to add a scale bar to the heat scatter plot, a few points showing potential changes to expression perhaps looked more apparent than they should. The new heat scatter has contour lines indicating deciles in the data and a count of promoters $< 1 \log_2$ fold change in relative expression ($>97\%$ for AYR/ARY libraries, $>93\%$ BYR library) and $< 0.5 \log_2$ fold change in relative expression ($>77\%$ AYR/ARY libraries and $>68\%$ BYR libraries). When looking at promoters in the middle 80% of DNA reads, the distribution narrows even further (new Figure S1H and below panel). While it is possible that there are some outliers that have meaningful effects on expression, this would not be the general rule. To address these outliers, UMIs would be required on DNA templates (not present in our experiment or in Vvedenskaya et al) and optimally, mRNA stabilities would need to be measured for all transcript isoforms taking account barcode effects if possible. These experiments would be beyond the scope of our study.

There is a weak negative correlation between the two variables in "ARY", and larger variations can be observed among variants with low TSS efficiency in "BYR". In addition, most TSS efficiency values are found between 20 to 80 in "ARY" in the figure, while the results in Fig 1D show that most TSS efficiency is below 20 in "ARY". I am wondering if different datasets were used.

*We apologize for confusion/miscommunication. Since ARY supports essentially no usage, the figure for review examined **+2 efficiency** for this library. Examination of +1 would essentially collapse the data on the x-axis (please see response figure below and new figure S1H that shows both +2 TSS and +1 TSS for the ARY library). The greater distribution of effects at low efficiencies in BYR is difficult to entangle from the fact that this is the largest library, with each promoter represented by less signal on average, and that this library has the majority of low efficiency TSSs because of the exclusion of -8A. We do filter by coefficient of variation (CV) of the three biological replicates yet the figure below shows that more promoters for BYR library are up against this threshold relative to the other libraries.*

CV v.s. total normalized DNA reads of biological replicates

More importantly, what I originally suggested was to use an alternative way to infer TSS efficiency (e.g., normalizing TSS-seq of +1 to DNA-seq read depth) to examine the robustness of the results, because the authors defined TSS efficiency as the relative distribution of TSS signals within a promoter, instead of absolute quantification of transcripts initiated from +1 TSS. Thus, it is necessary to account for expression differences in amount variants. If the normalized results are consistent with that of Fig 1D, it would further support its main conclusion.

We apologize for the misunderstanding here. The figure presented in response to review previously, and now present in the revised manuscript as Figure S1H is a version of this normalized analysis. The figure we show as S1H specifically indicates the number of promoters with $\log_2 -1$ to 1 and shows contours on the heat scatter plots representing deciles. ~70% of promoters are within $\log_2 -0.5$ to 0.5. Here, all RNA reads to all DNA reads are taken as a baseline ratio. Total reads from each promoter are then normalized to DNA template for that promoter and compared to the baseline ratio. Deviation from this ratio would be expected if a particular promoter sequence caused expression from a promoter to deviate from the average promoter output. What this is indicating is that $[(\text{Reads from } +1 \text{ plus all other reads})/(\text{DNA reads})]$ is essentially similar for all promoters. By plotting this ratio relative to the efficiency of +1 (AYR, BYR) or +2 (ARY), deviation from this trend based on efficiency at +1 (or +2) would be apparent, and we don't observe deviation. We have displayed the data below in three ways below for the middle 80% of promoters based on DNA reads. First, by mapping efficiency relative to normalized expression, results indicate the large majority of promoters are evenly expressed regardless of +1 or +2 efficiency; Second, by showing the high correlation (Pearson r shown) between TSS efficiency and TSS usage (%) relative to promoter usage; Third, by showing the high correlation between efficiency and TSS usage normalized to relative expression. We favor use of efficiency as we have calculated it, because this provides an internal normalization that will be more accurate than DNA template level, given PCR amplification effects. DNA template levels were determined without unique molecular identifiers (UMIs) and will have skew in distribution based on PCR duplicates. This skew is not present in our RNA samples as we have UMIs there. When DNA template levels are compressed to a range from 0-100%, this compression reduces the skew in DNA reads relative to efficiency calculated from RNA only, and shows a high correlation (second row below). The "sag" of efficiency relative to usage (second row of plots below) is expected because initiation anywhere downstream of the +1 will reduce calculated efficiency but have no effect on normalized +1 usage as percent of usage range observed.

2. For the second issue in my review, the authors again stated that "FD does not strongly affect upstream TSS", which is not well supported by the data. As results in Vvedenskaya et al 2015 also observed large differences in transcription efficiency among variants. Thus, this factor is not negligible. I still believe it is necessary to include additional analysis to show limited or no impact by different levels of transcriptions among variants.

The analysis presented above is the basis for our argument that there are not strong interactions between the FD and upstream TSSs. Our reasoning here is if there were interactions between the FD and upstream sequences, they hypothetically would be affected by the strength of the upstream sequence (as the FD is not variable). The total relative expression (normalized to DNA) for the majority of promoters is 1, meaning that when normalized to DNA the aggregate output of most promoters is approximately even, regardless of efficiency of the +1 TSS. This analysis suggests that the detected Pol II flux (Pol II that initiates at any start site in the promoter) is highly similar for each promoter (upstream TSSs+FD usage=same). Bacterial RNAP does not scan so there is no inherent value for a "flux detector" in those experiments. Our experiments suggest that overall promoter expression is relatively even across promoters, and this is consistent with the hypothesis that a low efficiency upstream site allows Pol II to continue scanning and initiate downstream. Therefore, there may be only marginal differences between a PIC scanning to the +1 position or continuing to scan to the FD.

3. Both reviewers 2 and 3 mentioned that this manuscript includes too many details and too many figures, which reduces its readability. I agreed with the two reviewers on this point. This issue has not been satisfactorily addressed in this revision, as the revised version is even longer (1180 lines vs 1106, excluding Figure & Figure Legends) and the number of main figures remains unchanged. The authors argued it is due to the addition of new data, there is not much improvement in this aspect.

Points 3 and 4 are addressed together below point 4.

4. As the revisions were not tracked in the manuscript, and specific revisions were not described in many point-to-point responses, it is very difficult to learn what exact revisions were made in the manuscript. For example, in responding to my comment about the

number of TSS sequences or promoter variants, the authors wrote "we have attempted to clarify...". It was unclear to me whether it had been clarified and what clarification was added.

In revision 1, we took the concerns of all reviewers seriously and extensively rewrote the manuscript. We have made additional changes that we now document to further address this concern. We note that this was specifically recognized by Reviewer 2 in their review of our Revision 1. For example, the section detailing Pol II mutant effects was shortened from ~108 lines to 30 lines, however 20 additional lines were added in results for the new data and 40 lines were added to methods for metabolomics analyses.

We have now provided a tracked changes version indicating the extent of changes from the original submission to Revision 1 and from Revision 1 to Revision 2.

I had to search the entire manuscript and compare it to the previous version to figure out what clarifications were added. I found that the authors added "Our three libraries comprise 81,920 promoter variants, allowing up to 983,040 individual TSS sequences" added to line 161. I understand how to get the number 81920 based on figure 1B, but it remains unclear to me how to get 983,040 individual TSS sequences. As the main work was to examine TSS efficiency changes in the manuscript. among "81,920 promoter variants", it is more appropriate to use this number in the abstract.

As each promoter variant can support initiation from multiple positions within and outside of the randomized region (not just the designated +1 region), each promoter variant allows multiple unique sequences to be analyzed for TSS efficiency, adding to the number of unique sequences we report for efficiency and include in our modeling. We have revised the statement to the following text (line 159-163):

Our three libraries comprise 81,920 promoter variants. Because each of these promoter variants were evaluated for TSS initiation efficiency at up to 12 positions within or adjacent to the randomized region, our assay allows for analysis of up to 983,040 distinct TSS sequences.

Reviewer #2:

Remarks to the Author:

The authors have done a good job in revising the manuscript in response to my comments and, in my opinion, responding to the comments from the other reviewers. The streamlined manuscript and revised figures are much easier to follow and it's much more apparent what are the mechanistic insights from the work. The authors should consider the following in a revised manuscript:

We thank the reviewer for their previous comments and the comments below.

1) Something seems wrong in the TSS sequence motif learned by the predictor in Fig 6C. In multiple prior figures, the critical importance of position -8 was demonstrated. However, in 6C the positions -8 (and -9) show only a weak bias for A compared with the strong sequence bias around the +1 position. The authors explanation as presented here is not convincing to me.

We think the confusion here is for the following reasons. First, the logo in Figure 2E is biased by the restriction of -9 to C (and potential additional upstream effects). -9C does appear to increase preference for -8A (see Figure 2F). Second, the sequence logo in Figure 5A is based on genomic TSSs which can be biased by evolution. Third, when we

include an interaction term between -8 and -9 positions, this has the result of diminishing each individual position because in essence, the model is allowing for one or the other to contribute. If we remove the interaction term and only consider -8 and -9 as independent contributors, the height of each position increases (see figure below).

2) The figure shown in the response to reviewer 1 (relative expression of promoter vs efficiency of major TSS) seems important. There is no point in showing this only to reviewers, so I think that this should be added to the supplementary figs.

Good point. We have added this figure to Fig S1H.

Reviewer #3:

Remarks to the Author:

This is a revised manuscript from Zhu et al describing "Pol II Master," a massively parallel assay for studying the nucleotide preferences of RNA pol II start sites. I thought the conclusions in the original manuscript were well supported by the data, and they remain well supported in the revision. I am satisfied with the authors' response to the question of over fitting in the model. I congratulate the author's on an interesting study and a clever technology.

Thank you very much for your positive response to our study.

Final Decision Letter:**Message** 8th Sep 2023

:

Dear Dr. Hegde,

We are now happy to accept your revised paper "EMC rectifies the topology of multipass membrane proteins" for publication as an Article in Nature Structural & Molecular Biology.

As soon as your article is published, you can generate your shareable link by entering the DOI of your article here: http://authors.springernature.com/share. Corresponding authors will also receive an automated email with the shareable link

Your paper will be published online soon after we receive proof corrections and will appear in print in the next available issue. You can find out your date of online publication by contacting the production team shortly after sending your proof corrections. Content is published online weekly on Mondays and Thursdays, and the embargo is set at 16:00 London time (GMT)/11:00 am US Eastern time (EST) on the day of publication. Now is the time to inform your Public Relations or Press Office about your paper, as they might be

interested in promoting its publication. This will allow them time to prepare an accurate and satisfactory press release. Include your manuscript tracking number (NSMB-A47639B) and our journal name, which they will need when they contact our press office.

About one week before your paper is published online, we shall be distributing a press release to news organizations worldwide, which may very well include details of your work. We are happy for your institution or funding agency to prepare its own press release, but it must mention the embargo date and Nature Structural & Molecular Biology. If you or your Press Office have any enquiries in the meantime, please contact press@nature.com.

Please note that *Nature Structural & Molecular Biology* is a Transformative Journal (TJ). Authors may publish their research with us through the traditional subscription access route or make their paper immediately open access through payment of an article-processing charge (APC). Authors will not be required to make a final decision about access to their article until it has been accepted. <https://www.springernature.com/gp/open-research/transformative-journals> Find out more about Transformative Journals

Sincerely,
Kat

Katarzyna Ciazynska
(she/her)
Associate Editor
Nature Structural & Molecular Biology
<https://orcid.org/0000-0002-9899-2428>
